# Functional Classification Under Local Differential Privacy with Model Reversal and Model Average

## Abstract

In the rapidly evolving landscape of technological development, privacy protection in data collection and storage demands heightened attention. While there has been notable research on releasing functional summaries of interest under Differential Privacy (DP), the area of learning models with functional observations under Local Differential Privacy (LDP) is still largely unexplored. This paper seeks to address this gap by implementing functional projection with a finite basis and introducing aggregation techniques that are well-suited for LDP, thereby contributing to the advancement of privacy-preserving methodologies in functional data analysis. Specifically, we propose algorithms for constructing functional classifiers designed for both single-server and heterogeneous multi-server environments under LDP. In single-server scenarios, we introduce an innovative allocation strategy where fewer samples are used for training multiple weak classifiers, while the majority are used to evaluate their performance. This enables the construction of a robust classifier with enhanced performance by model averaging. We also introduce a novel technique, *model reversal*, which effectively enhances the performance of weak classifiers. In multi-server contexts, we employ federated learning and enable each server to benefit from shared knowledge to improve the performance of each server's classifier. Experimental results demonstrate that our algorithms significantly boost the performance of functional classifiers under LDP.

## 1 Introduction

Advances in technologies enable us to collect and process data densely observed over temporal or spatial domains, which are termed *functional data* (Ramsay & Silverman, 2005; Ferraty, 2011; Horváth & Kokoszka, 2012), distinguishing them from traditional multivariate data. It is typically represented as curves, surfaces, or anything that varies over a continuum. Privacy preservation for functional data is an important issue, as the data may reveal individual characteristics or preferences through their temporal or spatial patterns. This is especially relevant in domains such as health informatics or behavioral science, where medical data of brain scans like DTI (Short et al., 2022) or fMRI (Logothetis, 2008) can show brain anatomy and activity, and smart devices like phones, watches, etc. can capture human activity data (Stisen et al., 2015). However, there are only a few works concerning privacy preservation within the realm of functional data.

Differential privacy (DP, Dwork et al., 2006) is a leading paradigm for privacy-preserving statistical analyses. It provides a rigorous and interpretable definition of data privacy, and limits the amount of information that attackers can infer from publicly released database queries. However, in some scenarios, when the data collector is untrusted or malicious, or when the data is highly sensitive or personal, DP may not be sufficient. In such cases, we need to consider local differential privacy (LDP, Kasiviswanathan et al., 2011) instead, which adds noise at the individual data level before centralization. It has been deployed by major technology companies like Apple (Differential Privacy Team, 2017), Google (Erlingsson et al., 2014), Windows (Ding et al., 2017). However, LDP often requires more noise to achieve the same privacy level as DP, which can lead to decreased utility of the data. Currently, two primary research directions under LDP are statistical queries and private learning(Yang et al., 2020). For statistical queries, where the aggregator aims to collect user data to address queries like frequency, mean, and range, there is plenty of research. Conversely, private

learning, which focuses on training models via specific machine learning algorithms, is comparatively less explored. The formal definition of LDP is as follows:

**Definition 1** ($\varepsilon$-LDP). *A randomized mechanism $\mathcal{M}$ satisfies $\varepsilon$-LDP, where $\varepsilon > 0$ is the privacy budget, if and only if for any pair of input values $u_1, u_2$ in the domain of $\mathcal{M}$ and any possible output $v$ of $\mathcal{M}$, it holds*

$$P(\mathcal{M}(u_1) = v) \leq e^{\varepsilon} P(\mathcal{M}(u_2) = v).$$

Private learning faces significant challenges, largely arising from data correlations and high dimensionality (Wang et al., 2020; Yang et al., 2020; Ye & Hu, 2020). Firstly, challenges arise in preserving correlations among multiple features (Wang et al., 2019b) and between features and their corresponding labels (Yilmaz et al., 2019). Second, when collecting $d$-dimensional data under LDP, common approaches involve either distributing the privacy budget $\varepsilon$ across the $d$ dimensions and reporting every dimension with $\varepsilon/d$-LDP or allowing users to randomly select a single dimension to report with $\varepsilon$-LDP (Nguyên et al., 2016; Arcolezi et al., 2021; 2022). A challenge emerges as an increase in $d$ leads to a rapid decay of the privacy budget and a rise in the noise scale.

In this paper, we propose two novel algorithms to construct functional classifiers under LDP, suitable for single-server and heterogeneous multi-server settings, respectively. Our contributions are summarized as follows:

- **Present a process of constructing functional classifiers under LDP**. We develop and offer theoretical analysis on a projection-based functional classifier, and measure the information loss in classification induced by projection. To the best of our knowledge, this is the first work that models functional data under LDP.

- **Introduce a novel technique *model reversal***. It improves the performance of weak classifiers under LDP by inverting their prediction direction ($-1\times$ coefficient estimate) when their accuracy is below a certain threshold. Given a classifier's accuracy rate distribution, we measure the improvement that model reversal can bring to this classifier.

- **Propose a model average algorithm tailored for LDP**. It includes our idea of allocating a larger proportion of clients to evaluate the performance of weak classifiers instead of training them. It builds upon our methods of evaluating the performance of weak classifiers under LDP, and assigning more suitable weights to these classifiers based on the evaluation.

- **Extend to heterogeneous multi-server settings**. We propose a federated learning approach under LDP, where each server benefits from the shared knowledge of others.

## 2 RELATED WORK

**Supervised Learning Under LDP.** Given challenges arising from data correlations and high dimensionality, existing research on supervised learning under LDP is limited. Both Yilmaz et al. (2019) and Xue et al. (2019) aimed to train a Naïve Bayes classifier under LDP while preserving the correlation between feature values and class labels. However, their methods did not demonstrate distinct advantages and suffered from high variance and low utility when the number of features or the size of the input domain is large. Alternatively, some research focuses on empirical risk minimization (ERM), treating the learning process as an optimization problem solved through defining a series of objective functions. Wang et al. (2019a) constructed a class of machine learning models under LDP that can be expressed as ERM, and solved by stochastic gradient descent (SGD). To address the high dimensionality issue, Liu et al. (2020) privately selected the top-$k$ dimensions according to their contributions in each iteration of federated SGD. Additionally, Sun et al. (2020), which trained deep learning models in federated learning (Konečnỳ et al., 2016) under LDP, proposed data perturbation with adaptive range and a parameter shuffling mechanism to mitigate the privacy degradation caused by high dimensionality. However, to the best of our knowledge, no existing research has explored modeling with functional data, which has infinite dimensionality, within the framework of LDP.

**Functional Data Projection.** Recent research has begun to employ projection to reduce the dimensionality of functional data (Horváth & Rice, 2015; Pomann et al., 2016; Kraus & Stefanucci, 2019). These methods are facilitated by the inherent property of functional data, which allows it to

be represented by a linear combination of a set of basis functions. Specifically, to test the equality of two-sample mean functions, Meléndez et al. (2021) and Górecki & Smaga (2015) developed test statistics using projections with finite random functions and data-independent basis functions, respectively. For the functional classification problem, Kraus & Stefanucci (2019) and Sang et al. (2022) constructed classifiers based on projections with estimated finite functions.

**Functional Data Under DP and LDP.** Previous DP research primarily concentrated on scenarios where a query's output is a vector. Hall et al. (2013) was a pioneer in providing a framework for achieving $(\varepsilon, \delta)$-DP with infinite dimensional functional objects, yet it focused on a finite grid of evaluation points. Subsequently, Mirshani et al. (2019); Reimherr & Awan (2019); Jiang et al. (2023) conducted further extensive work around $(\varepsilon, \delta)$-DP. Regarding $\varepsilon$-DP, a series of studies have been conducted that utilize finite-dimensional representations (Wang et al., 2013; Alda & Rubinstein, 2017). More recently, Lin & Reimherr (2023) introduced a novel mechanism for releasing $\varepsilon$-DP functional summaries, termed the Independent Component Laplace Process, which relaxes assumptions on data trajectories and enhances data utility. However, releasing functional summaries under DP and collecting functional observations under LDP are significantly different tasks. Under DP, the server can access all the original data and achieve privacy based on the knowledge of covariance function and sensitivity of functional objects. Under LDP, however, each functional object must be noised before being sent to the server. Moreover, DP focuses on the quality of the functional summaries, while LDP emphasizes the performance of the models trained on the noised functional observations from the clients. Thus, using finite basis projection for privacy preservation may have less impact under LDP than under DP.

## 3 SINGLE SERVER WITH MODEL REVERSAL AND MODEL AVERAGE

**Problem Setup**. Suppose there is a server with $N$ clients. Each client has a square-integrable functional covariate $x(t)$, which is defined over a compact domain $\mathcal{I}$, and the corresponding binary label $y$. Without loss of generality, we assume that $y \in \{0, 1\}$ and $\mathcal{I} = [0, 1]$. The primary goal is to construct a classifier $f$ based on $\{(x_i, y_i)\}_{i=1}^{N}$ under local differential privacy (LDP), where

$$f(x) = \alpha + \int x(t)\beta(t)dt,$$

and for a new sample with functional covariate $x(t)$, we predict its label $y$ as $\hat{y}(x) = I(f(x) > 0)$.

As pointed by Yang et al. (2020), LDP algorithms typically involve four steps: encoding, perturbation, aggregation, and estimation. In the first two steps, each client encodes his original value according to the predefined coding scheme, and perturbs encoded value by the randomized algorithm that achieves local differential privacy. Then, in the last two steps, the server collects all the perturbed value from clients and estimates the query result. In the following, we introduce the processes of estimating the functional classifier $f$ under LDP. The overall framework of our algorithm is summarized in Algorithm 1.

### 3.1 ENCODING AND PERTURBATION

For a client with functional covariate $x(t)$ and label $y$, the data he reports is transformed through the following steps.

**Dimensionality Reduction**. The infinite dimensionality of functional data is a significant challenge in functional data analysis, especially under LDP. To address this challenge, we consider mapping $x(t)$, onto a finite-dimensional functional space, spanned by functions $\phi_1, \ldots, \phi_d$, resulting in a low-dimensional vector $\boldsymbol{z} \in \mathbb{R}^d$. This vector is then used as a substitute for $x(t)$ when building the subsequent classifier.

Specifically, the projection functions, $\Phi = (\phi_1, \ldots, \phi_d)$, are prespecified by the server, and generally can be taken as B-Spline basis, Fourier basis, etc. The mapping process is equivalent to representing $x(t)$ by the truncated basis $\Phi$, i.e.,

$$x(t) = \sum_{k=1}^{d} z_k \phi_k(t) + \xi(t), \tag{1}$$

where $\boldsymbol{z} = (z_1, \ldots, z_d)^\top$ is the coefficient vector of $\Phi$, and $\xi(t)$ is the residual function that can't be represented by $\Phi$. Regarding the expressive capacity of a finite basis, we propose that employing a finite $d$ is feasible, considering the following three aspects:

1. **Functional data fitting**: In practice, functional observations are vectors observed at different time points, i.e., $\boldsymbol{x} = (x(t_1), \ldots, x(t_T))^\top$. When fitting with B-Spline basis of order $m$ ($m = 4$ for cubic spline), the estimation risk decays at the optimal rate of $T^{-2m/(2m+1)}$ as the number of knots grows like $T^{1/(2m+1)}$ (Eubank, 1999), which indicates a relatively slow rate of increase. Note that the number of basis $d$ equals the order $m$ plus the number of knots, so finite B-Spline basis can achieve an effective fitting;

2. **Performance of projection-based classifiers**: Lemma 1 in Appendix B.1 demonstrates that, under the assumption that $x(t)$ is from a Gaussian process, classifiers based on projection with finite basis can achieve the minimum misclassification rate;

3. **Functional projection under LDP**: Using a finite basis helps to capture population-level patterns while avoiding overfitting individual differences. Additionally, replacing $x(t)$ with $\boldsymbol{z}$ significantly improves communication efficiency.

In Appendix B.1, we provide further discussion on the projection-based functional classification and measure the information loss induced by projection. And we compare the results of $d$ at different values in Appendix A.2.

**Rescaling**. Before adding noise, to bound the sensitivity, clipping and truncation are commonly adopted to rescale the domain of each numeric feature. Here we introduce two types of transformation of rescaling $\boldsymbol{z} \in \mathbb{R}^d$ into $\boldsymbol{z}^* = (z_1^*, \ldots, z_d^*)^\top \in [-1, 1]^d$, where for $k = 1, \ldots, d$,

$$\text{(Tanh Transformation)} \quad z_k^* = \tanh(z_k),$$
$$\text{(Min-Max Transformation)} \quad z_k^* = \frac{z_k - \min_{k'} |z_{k'}|}{\max_{k'} |z_{k'}| - \min_{k'} |z_{k'}|}.$$

The Tanh transformation introduces non-linearity, whereas the Min-Max transformation maintains the relative relationships between the original data points. For different types of $\Phi$, the two transformations are suitable for distinct scenarios. For instance, when employing the B-Spline basis as $\Phi$, the Tanh transformation is generally more appropriate for cases where functions $x(t)$ in different classes exhibit significant magnitude or value differences. Conversely, the Min-Max transformation is more fitting for situations where functions $x(t)$ in different classes exhibit distinct shapes or trends within the $[0, 1]$ range. In Appendix A.2, we compare the performance of these two transformations.

**Perturbation**. Based on Laplace mechanism (Dwork et al., 2006), the rescaled $\boldsymbol{z}^*$ can be perturbed by adding randomized Laplace noise, i.e., reporting $\boldsymbol{z}' = (z_1', \ldots, z_d')$ with

$$z_k' = z_k^* + \mathrm{e}_k, \tag{2}$$

where $\{\mathrm{e}_k\}_{k=1}^d$ are independently drawn from a Laplace distribution with scale parameter $\lambda = d\Delta/\varepsilon_1$ and $\varepsilon_1 < \varepsilon$. Please note that $\Delta$ is the sensitivity of $z_k^*$ and $\Delta = 2$ since $z_k^* \in [-1, 1]$, and the privacy budget for each dimension is $\varepsilon_1/d$. Additionally, through randomized response Warner (1965), the label $y$ can be perturbed by reporting $y'$, where

$$P(y' = y) = e^{\varepsilon_2}/(1 + e^{\varepsilon_2}), \tag{3}$$

and $\varepsilon_2 = \varepsilon - \varepsilon_1$. Thus, a client with observation $(x(t), y)$ only needs to report $(\boldsymbol{z}', y')$. In this paper, we take $\varepsilon_2 = \varepsilon/(d + 1) = \varepsilon_1/d$, while different ways of allocating the privacy budget warrant further exploration. The following Theorem 1 demonstrates that the encoding and perturbation process adheres to $\varepsilon$-LDP.

**Theorem 1.** *Let $\mathcal{M}_1$ be the privacy mechanism that takes $(x(t), y)$ as input and outputs $(\boldsymbol{z}', y')$ as described above. Then $\mathcal{M}_1$ satisfies $\varepsilon$-local differential privacy.*

### 3.2 Aggregation and Estimation

After collecting all the perturbed values $\{(\boldsymbol{z}_i', y_i')\}_{i=1}^N$ from clients, we provide two ways of building functional classifiers based on the collected data.

**Method I**. It's straightforward to first build a vector classifier with the $d$-dimensional predictor $z_i'$ by traditional methods, such as logistic regression and support vector machine (SVM). Record the estimated intercept and coefficient as $\hat{\alpha} \in \mathbb{R}$ and $\hat{b} \in \mathbb{R}^d$, respectively. Then the slope function $\beta(t)$ can be estimated by $\widehat{\beta}(t) = \sum_{k=1}^d \hat{b}_k \phi_k(t)$, and the corresponding functional classifier is

$$\hat{f}(x) = \hat{\alpha} + \int x(t)\widehat{\beta}(t)dt.$$

**Method II**. Before building the classifier, we can first perform curve recovery, i.e., treating $x_i'(t) = \sum_{k=1}^d z_{i,k}' \phi_k(t)$ as the perturbed functional covariate, where $z_{i,k}'$ represents the $k$th element of $z_i'$. Then, the classifier can be obtained using the functional conjugate gradient algorithm (CG, Kraus & Stefanucci, 2019) or the functional distance weighted discrimination (DWD, Sang et al., 2022).

With these two ways of constructing classifiers, in the Appendix A.2, we illustrate the effects of dimensionality reduction, rescaling, and perturbation on the misclassification rate of the classifiers. Experiment results indicate that the projection of function data and rescaling of coefficient vectors have a small impact on the classifier's performance. And the performance of two types of transformations is similar in our context. Moreover, as the privacy budget $\varepsilon \to 0$, the misclassification rates of the classifiers based on the perturbed data tend to 50%.

## 3.3 MODEL AVERAGE

While LDP ensures strong privacy guarantee, it introduces significant noise. Modeling on perturbed data often leads to poor performance. To address this, we propose to leverage model averaging to enhance classifier performance. Therefore, to assign appropriate weights for weak classifiers, we partition $N$ clients into two sets: a training set $\mathcal{D}$ with size $N_0$ for constructing weak classifiers and a validation set $\mathcal{D}_{\text{valid}}$ with size $N_1$ for evaluating their performance, where $N_0 + N_1 = N$.

**Training**. The server collects the perturbed value $\{(z_i', y_i')\}_{i \in \mathcal{D}}$ from the clients in the training set. Based on this perturbed training data, multiple weak classifiers can be constructed by random sampling. Specifically, we build $B$ weak classifiers, each based on a random sample of $n_0(< N_0)$ training instances. Denote the classifiers as $\{f^{(b)}\}_{b=1}^B$, and their corresponding coefficients as $\{(\hat{\alpha}^{(b)}, \widehat{\beta}^{(b)}(t))\}_{b=1}^B$. Note that the privacy guarantee is not affected by building multiple classifiers based on the same dataset, since each client reports the perturbed value only once.

**Validation**. For the clients in the validation set, instead of collecting their perturbed observations, we obtain their perturbed evaluations of a classifier's performance. Specifically, for the given $B$ classifiers, we split the validation set into $B$ subsets, denoted as $\mathcal{D}_{\text{valid}}^{(b)}, b = 1, \ldots, B$, and evaluate the performance of the classifier $f^{(b)}$ based on the reports of the clients in the subset $\mathcal{D}_{\text{valid}}^{(b)}$. Each client in $\mathcal{D}_{\text{valid}}^{(b)}$ with observation $(x_i(t), y_i)$ calculates $\hat{y}_i = I(f^{(b)}(x_i) > 0)$ and $r_i = I(\hat{y}_i = y_i)$, and reports $r_i'$ with $P(r_i' = r_i) = q$ and $q \in (1/2, 1)$. In Theorem 2, we demonstrate that the validation process adheres to LDP, and provide an unbiased estimate of the classifier's accuracy.

**Theorem 2.** *Let $\mathcal{M}_2$ be the privacy mechanism that takes $(x(t), y)$ as input and outputs $r'$ as described above. Then for $q = e^{\varepsilon_v}/(1 + e^{\varepsilon_v})$, $\mathcal{M}_2$ satisfies $\varepsilon_v$-local differential privacy. Furthermore, let $r^{(b)}$ be the classification accuracy of the classifier $f^{(b)}$, $n_1^{(b)} = |\mathcal{D}_{\text{valid}}^{(b)}|$, and*

$$\tilde{r}^{(b)} = \frac{\hat{r}^{(b)} + q - 1}{2q - 1} \text{ with } \hat{r}^{(b)} = \frac{\sum_{i \in \mathcal{D}_{\text{valid}}^{(b)}} r_i'}{n_1^{(b)}}. \tag{4}$$

*Then $\mathbb{E}(\tilde{r}^{(b)}) = r^{(b)}$ and $Var(\tilde{r}^{(b)}) \leq ((e^{\varepsilon_v} + 1)/(e^{\varepsilon_v} - 1))^2/(4n_1^{(b)})$.*

From Theorem 2, it's evident that as the sample size for classifier evaluation increases, our estimate under LDP tends to approximate the classifier's true accuracy. This inspired us to allocate a larger proportion of clients for evaluating the performance of weak classifiers rather than for training them. This idea is particularly effective under substantial noise interference, where increasing the training sample size may yield limited performance gains. Conversely, expanding the validation sample size enhances the accuracy of assessments, significantly aiding in identifying the most effective weak classifiers and contributing to the subsequent development of a superior ensemble classifier.

In the single-server setting, since each client is used to evaluate just one classifier, we set $\varepsilon_v = \varepsilon$ to achieve $\varepsilon$-LDP. If one client is used to evaluate multiple classifiers, the privacy budget must be shared among them, which will further lower the accuracy of the evaluation.

**Model Reversal (MR)**. To optimize the performance of weak classifiers, we propose the *model reversal* technique. For a classifier with estimated accuracy $\tilde{r}^{(b)} < 50\%$, we invert the sign of its estimated coefficients, transforming $(\hat{\alpha}^{(b)}, \widehat{\beta}^{(b)}(t))$ to $(-\hat{\alpha}^{(b)}, -\widehat{\beta}^{(b)}(t))$, which results in a classifier with an enhanced estimated accuracy $1 - \tilde{r}^{(b)} > 50\%$. This approach is underpinned by the principle that, in classification tasks, the angle between the estimated and true coefficient values is paramount, unlike in regression where the emphasis is on minimizing their $L_2$ distance. The improvement it provides relies on our accurate evaluation of the performance of classifiers in Theorem 2.

**Theorem 3.** *For a classifier $f_\varepsilon$ that adheres to $\varepsilon$-LDP, let's denote its classification accuracy rate as $r(x) = P(sign(f_\varepsilon(x)) = y|x)$. Additionally, let $r_\delta$ represent the potential enhancement in classification accuracy that could be achieved for $f_\varepsilon$ through the application of model reversal. Then, if $r(x) = r_0$ for all $x(t)$, we have $r_\delta = \max\{1 - 2r_0, 0\}$. Otherwise, let's denote the distribution of classification accuracy rate as $p_\varepsilon(r) = P(x(t) \in A_{\varepsilon,r})$ with $r \in [0,1]$, where $A_{\varepsilon,r} = \{x(t)|r(x) = r\}$. Then we have*

$$\mathbb{E}(r_\delta) = \int_0^1 p_\varepsilon(r) \max\{1 - 2r, 0\} dr = \int_0^{1/2} p_\varepsilon(r)(1 - 2r) dr.$$

Theorem 3 measures the enhancement that model reversal can bring to the classifier $f_\varepsilon$, which depends on the privacy budget $\varepsilon$ and the distribution $p_\varepsilon(r)$. Simulation results in Figure 1 indicate that different classifiers vary in their sensitivity to noise and their distribution $p_\varepsilon(r)$. Theoretical analysis of the distribution $p_\varepsilon(r)$ for different classifiers is beyond the scope of this paper.

**Model Selection and Model Average (MA)**. Given the $B$ weak classifiers, it's essential to establish a criterion for selecting the top-performing classifiers and then effectively combine them and get a more robust model. For a specified cutoff value $r_0 \in (0.5, 1)$, we assign weight $w_b$ to the classifier $f^{(b)}$, where

$$w_b = \frac{\max(\tilde{r}^{(b)} - r_0, 0)}{\sum_{b=1}^B \max(\tilde{r}^{(b)} - r_0, 0)}. \tag{5}$$

Clearly, classifiers with $\tilde{r}^{(b)} \le r_0$ are excluded. And our final estimated classifier is

$$f^*(x) = \hat{\alpha}^* + \int x(t)\widehat{\beta}^*(t)dt \text{ with } \hat{\alpha}^* = \sum_{b=1}^B w_b \hat{\alpha}^{(b)}, \widehat{\beta}^*(t) = \sum_{b=1}^B w_b \widehat{\beta}^{(b)}(t). \tag{6}$$

In the experiments, we demonstrate the improvements in classification accuracy brought about by model reversal and model average.

**Sample Size Balancing**. When the total sample size $N$ is substantial, it stands to reason that increasing both the number of samples used to the estimation and evaluation of each classifier and the total number of classifiers can be advantageous. Yet, in practical, for a given $N$, it is imperative to deliberate on specifying the values of $N_0$ and $B$. While larger values of $N_0$ and $B$ assist in acquiring more effective weak classifiers, smaller values of $N_0$ and $B$ are preferred to ensure adequate sample sizes for each validation subset. In the Appendix A.3, we evaluate classifier performance across varying values of the parameters $N_0, N_1, n_0$ and $B$, and further discuss sample allocation.

## 4    MULTI-SERVER WITH FEDERATED LEARNING

In the previous section, we introduced the construction of classifiers on a single server through Algorithm 1. In real-world scenarios, multi-server environments are common. Thus, federated learning offers a promising approach. Suppose there are $K$ servers in total, each possessing a classifier, denoted by $f_k^*(x)$, trained on its own data, and these servers mutually exchange their models. In this section, we illustrate how each server can enhance classifier performance by effectively integrating classifiers from various heterogeneous servers under LDP. The details are outlined in Algorithm 2 in Appendix A.6.

---

**Algorithm 1** Functional Classification under LDP with MRMA

---

1: **procedure** SERVER($\Phi, N_0, N_1, n_0, B, \varepsilon, \varepsilon_v, r_0$)
      Divide clients into a training set $\mathcal{D}$ and a validation set $\mathcal{D}_{\text{valid}}$ with $|\mathcal{D}| = N_0, |\mathcal{D}_{\text{valid}}| = N_1$, and spilt clients in the validation set into $B$ subsets, $\mathcal{D}^{(b)}_{\text{valid}}, b = 1, \ldots, B$.
2:     **for** each client in the training set **do**
3:         $(\boldsymbol{z}'_i, y'_i) \leftarrow$ **ClientTrain**$(\Phi, \varepsilon)$.
4:     **end for**
5:     **for** $b = 1, \ldots, B$ **do**
6:         Randomly draw $n_0$ samples from $\{(\boldsymbol{z}'_i, y'_i)\}_{i \in \mathcal{D}}$ without replacement and denote as $\mathcal{D}^{(b)}$.
7:         Build a classifier $f^{(b)}$ based on $\{(\boldsymbol{z}'_i, y'_i)\}_{i \in \mathcal{D}^{(b)}}$.       ▷ See Section 3.2
8:         **for** each client in the validation set $\mathcal{D}^{(b)}_{\text{valid}}$ **do**
9:             $r'_i \leftarrow$ **ClientValid**$(f^{(b)}, \varepsilon_v)$
10:         **end for**
11:         Estimate $\tilde{r}^{(b)}$ by Equation 4.
12:         **if** $\tilde{r}^{(b)} < 50\%$ **then**
13:             $f^{(b)} \leftarrow -f^{(b)}$       ▷ Model Reversal
14:         **end if**
15:     **end for**
      Estimate $\boldsymbol{w} = (w_1, \ldots, w_B)^\top$ by Equation 5 with cutoff value $r_0$.
      **return** The final estimated classifier $f^*$ by Equation 6.       ▷ Model Average
16: **end procedure**

1: **procedure** CLIENTTRAIN$(\Phi, \varepsilon)$
2:     Estimate $\boldsymbol{z}$ through the functional regression in Equation 1.
3:     Rescale $\boldsymbol{z}$ into $\boldsymbol{z}^* \in [-1, 1]^d$ by Tanh or Min-Max transformation.
4:     Generate $\boldsymbol{z}', y'$ by Equation 2 and Equation 3, respectively.
      **return** $(\boldsymbol{z}', y')$
5: **end procedure**

1: **procedure** CLIENTVALID$(f^{(b)}, \varepsilon_v)$
2:     Calculate $\hat{y} = I(f^{(b)}(x) > 0)$ and $r = I(\hat{y} = y)$.
3:     Generate $r'$ by $P(r' = r) = e^{\varepsilon_v}/(1 + e^{\varepsilon_v})$.
      **return** $r'$
4: **end procedure**

---

**Perturbation**. When a server possesses ample data, the validation set can be partitioned into $B + K$ subsets initially, where the latter $K$ subsets are used to evaluate the performance of $\{f^*_k(x)\}^K_{k=1}$ and clients report the perturbed evaluations under $\epsilon$-LDP as in Section 3.1. In scenarios where a server's data is limited, one may consider the iterative use of the validation set. Initially, the validation set is partitioned into $B$ subsets, and clients report the perturbed evaluations of the classifiers constructed from their server under $\varepsilon_v = \epsilon/2$-LDP. Subsequently, the validation set is divided into $K$ subsets, and clients report the perturbed evaluations of classifiers from different servers under $\varepsilon^*_v = \epsilon/2$-LDP.

**Federated Learning**. Each server, upon receiving the $K$ classifiers and obtaining their performance evaluations, can refer to Section 3.2 to undertake both model reversal and model average, where we record the cutoff value used here as $r^*_0$ and the obtained classifiers as $\{f^\dagger_k\}^K_{k=1}$. Considering the heterogeneity across servers, model averaging in this context is similar to transfer learning (Olivas et al., 2009). To optimize this process, it is crucial to prevent the transmission of information that is either irrelevant or detrimental, thereby avoiding the negative transfer effect (Li et al., 2022).

## 5 EXPERIMENTS

In this section, we demonstrate the improvements in classification accuracy brought about by model reversal and model average. The data generation process is given in Appendix A.1. Assuming the server has a total of $N = 3000$ clients, we allocate $N_0 = 500$ for training and the remaining $N_1 = 2500$ for validation. To construct classifiers, we sequentially draw $n_0 = 50$ samples from the training dataset without replacement, repeating this procedure $B = 50$ times. At the same

time, we partition the validation set into $B = 50$ subsets of equal size, enabling us to evaluate the classification accuracy of each classifier using $n_1 = 50$ distinct samples. Both the training and validation of classifiers follow the data processing and privacy protection mechanisms proposed in Section 3. To assess the performance of the classifiers, we randomly generate a testing dataset comprising 500 samples during each trial, repeating this procedure 500 times.

Figure 1 showcases the misclassification rates, along with their corresponding error bars, for various classifiers across different $\varepsilon$ levels. In this figure, "Weak" denotes the average misclassification rate of $B = 50$ weak classifiers obtained through sampling. "MR" represents the average misclassification rate of $B$ weak classifiers after model reversal under LDP. "MA" signifies the results when using model average on weak classifiers with cutoff value $r = 0.4$, while "MRMA" illustrates the results of applying both model reversal and model averaging. To compare with classic aggregation methods, we train $B$ weak classifiers under LDP. Each classifier is trained with $N/B$ instances from the combined training and validation set. We then obtain the results through majority voting and averaging with equal weight, denoted as "Voting" and "Averaging", respectively. And "All data" denotes the classifier trained with $N$ clients directly.

The classifier "All data", even if it is trained with 3000 clients, shows almost no improvement over the classifier "Weak" when $\varepsilon$ is small (indicating substantial noise interference). And classifiers "Voting" and "Averaging" also perform similarly. However, our proposed techniques, both model reversal and model average, significantly improve the performance of all types of weak classifiers. And MRMA further enhances the performance of SVM and CG classifiers substantially. Figure 7 in Appendix A.4 demonstrates that even when allocating more clients for training weak classifiers, MR and MA can still significantly enhance the classifiers' performance. For further discussions regarding distinctions among different types of classifiers, please refer to Appendix A.2.

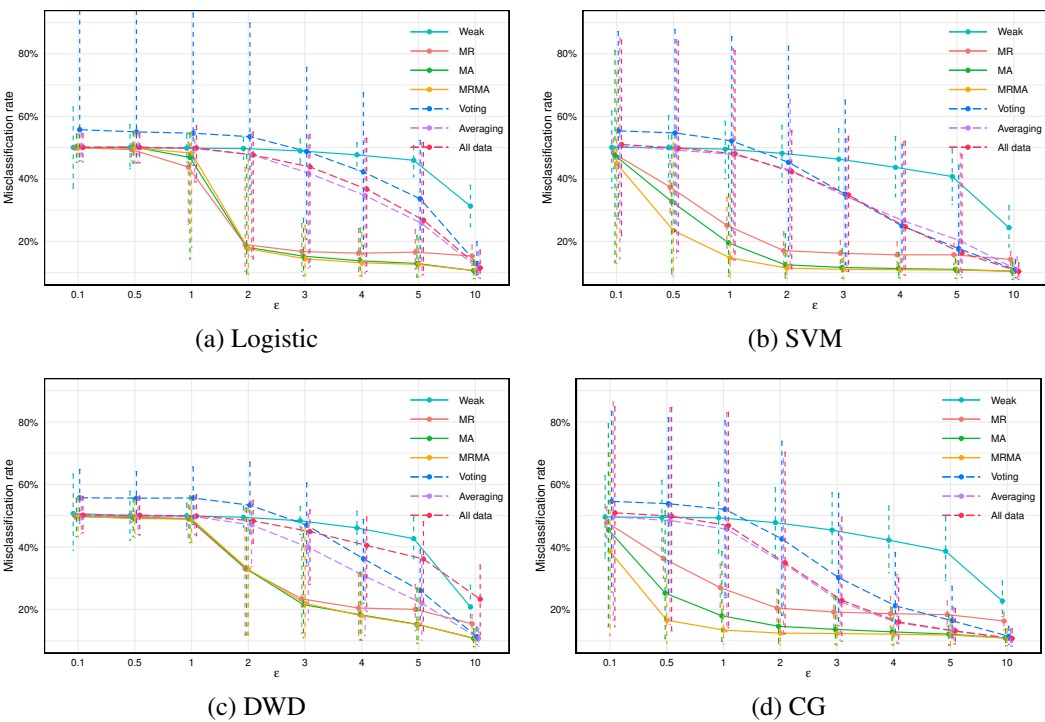

Figure 1: The misclassification rates of four types of classifiers with a single server under $\varepsilon$-LDP.

## 6 REAL APPLICATION

In this section, we employ a phonemes dataset derived from the TIMIT Acoustic-Phonetic Continuous Speech Corpus (Garofolo, 1993). Speech frames in this dataset are extracted from the continuous speech of 50 male speakers, and a log-periodogram is constructed from recordings available

at different equispaced frequencies for the five phonemes: "sh" as in "she", "iy" as in "she", "dcl" as in "dark", "aa" as in "dark", and "ao" as in "water". The log-periodogram data can be viewed as functions of frequency, rendering them as functional data, and can be fitted using a Fourier basis. The richness and potential sensitivity of speech data, capable of revealing unique aspects of an individual's identity such as accent, speech patterns, and even native language or regional background, necessitate the application of LDP.

In our study, we focus on classifying the phonemes "sh" and "iy", represented by 1163 and 872 log-periodograms, respectively, each of length 256 and with known class memberships. In Figure 8 in Appendix A.5, we visualize 200 randomly selected functional observations from each group, and the curves reconstructed after finite basis projection and transformation. To construct classifiers, we adopt a randomized approach to split the dataset into a testing set (535 instances), a training set (300 instances), and a validation set (1200 instances), and $B = 24$ weak classifiers are learned. This entire procedure is replicated 500 times to ensure robustness and reliability of our results.

Figure 2 presents the misclassification rates of classifiers. The results clearly indicate that classifiers based on model reversal and model averaging show significant improvement compared to other classifiers, especially when $\varepsilon$ is small.

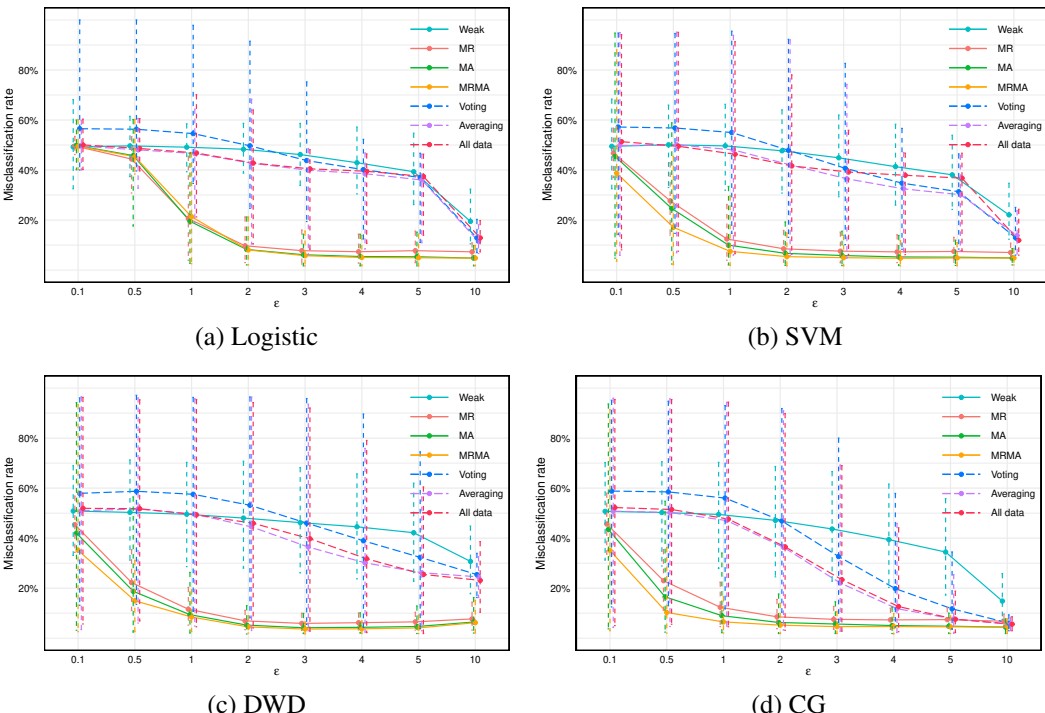

(a) Logistic  (b) SVM

(c) DWD  (d) CG

Figure 2: The misclassification rates of four types of classifiers with a single server under $\varepsilon$-LDP.

## 7 CONCLUSION

In this paper, we delved into functional data classification under local differential privacy, a domain that remains relatively underexplored. We introduced innovative algorithms designed for both single and multi-server environments. Notably, our allocation strategy prioritized a substantial proportion of clients for performance evaluation over training, paving the way for potential advancements. Further advancements include the *model reversal* technique, which enhanced weak classifier performance by reversing prediction outcomes, and the adoption of model averaging, which effectively combined weak classifiers. We also applied federated learning in a multi-server context, allowing servers to mutually benefit from shared knowledge. Experimental results demonstrated that our algorithms brought significant improvements to classifiers under LDP constraints. Furthermore, the methodologies we introduced hold promise for broader applications and future exploration.

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

# A  ADDITIONAL RESULTS

This section provides additional results on our experiments.

## A.1  DATA GENERATION PROCESS IN EXPERIMENTS

In each experimental trial, the functional covariate $X(\cdot)$ is generated by $X(t) = \sum_{j=1}^{50} \xi_j \, \zeta_j \phi_j(t)$ for $t \in [0, 1]$, where $\xi_j$'s are independently drawn from a uniform distribution on $(-\sqrt{3}, \sqrt{3})$, $\zeta_j = (-1)^{j+1} j^{-1}, j = 1, \ldots, 50$, $\phi_1(t) = 1$ and $\phi_j(t) = \sqrt{2} \cos((j-1)\pi t)$ with $j \geq 2$. The binary response variable $Y$, taking values 1 or 0, is generated using the following logistic model:

$$f(X) = \alpha_0 + \int_0^1 X(t)\beta(t)dt, \quad \Pr(Y = 1) = \frac{\exp\{f(X)\}}{1 + \exp\{f(X)\}},$$

where $\alpha_0 = 0.1$, $\beta(t)$ is the slope function, and $f(X)$ is referred to as the classification function. And we generate data for a server using the slope function $\beta(t) = \sum_{j=1}^{50} 4(-1)^{j+1} j^{-2} \phi_j(t)$.

Results in Appendix A.2 show that the performance with $d = 4, 5, 6$ cubic B-Spline is comparable, and the performance based on Tanh or Min-Max transformation is close. Thus in Section 5, we present results with $d = 4$, which introduces less noise during perturbation, and employ the tanh transformation.

## A.2  ENCODING AND PERTURBATION

In this section, we discuss the effects of dimensionality reduction, rescaling, and perturbation, as introduced in Section 3.1, on the misclassification rates of different types of classifiers.

To investigate the impacts of dimensionality reduction and rescaling, we generate data based on the model described in Section 5, and the sample sizes of the training and testing datasets are 50 and 500, respectively. During the dimensionality reduction, cubic B-spline with equidistant knots are employed, and we apply the methods for varying numbers of basis functions, $d = 4, 5, 6$. Table 1 showcases the misclassification rates of classifiers based on the actual data (INI), coefficients obtained after dimensionality reduction (Coefs), and coefficients rescaled by either the Tanh (Tanh) or Min-Max (MM) transformation. This is based on the results from 500 repeated experiments.

Table 1: The misclassification rate of classifiers based on actual data (INI), coefficients obtained after dimensionality reduction (Coefs), and coefficients rescaled by either the Tanh (Tanh) or Min-Max (MM) transformation

|          | INI   | $d = 4$ | | | $d = 5$ | | | $d = 6$ | | |
|          |       | Coefs | Tanh  | MM    | Coefs | Tanh  | MM    | Coefs | Tanh  | MM    |
|----------|-------|-------|-------|-------|-------|-------|-------|-------|-------|-------|
| Logistic | -     | 17.81 | 12.32 | 11.68 | 20.66 | 13.55 | 12.81 | 23.19 | 15.21 | 13.86 |
| SVM      | -     | 10.85 | 11.20 | 11.03 | 11.29 | 11.67 | 11.27 | 11.69 | 12.07 | 11.48 |
| DWD      | 11.01 | 11.02 | 10.64 | 10.61 | 11.01 | 10.60 | 10.58 | 11.01 | 10.61 | 10.59 |
| CG       | 15.42 | 11.02 | 11.47 | 11.31 | 11.34 | 11.86 | 11.43 | 11.65 | 12.00 | 11.64 |

Table 1 shows that there are slight variations in the performance of different types of classifiers. Overall, classifiers with $d = 4, 5, 6$ exhibit comparable results, and the impact of dimension reduction and rescaling on classifier performance is small. Also, the performances based on Tanh and Min-Max transformations are very similar. Furthermore, the CG classifier based on actual data and the logistic classifier based on coefficients obtained after dimensionality reduction perform relatively poorer. This may be related to the inherent characteristics of the classifiers, which is beyond the scope of this paper. Our primary focus is on the changes in classifier performance before and after consider LDP and the improvements brought about by different techniques.

To further explore the impacts of perturbation, we generate data in accordance with the model in Section 5, and both the training and testing dataset sample sizes are 500. Table 1 demonstrates

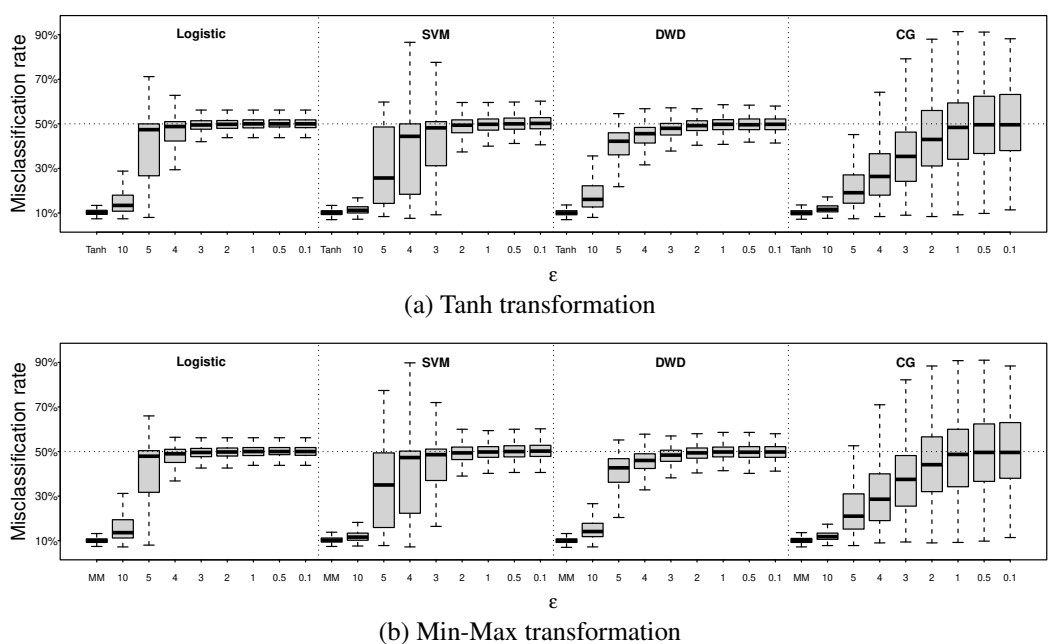

Figure 3: The boxplot of the misclassification rates of classifiers with Tanh and Min-Max transformations under $\varepsilon$-LDP.

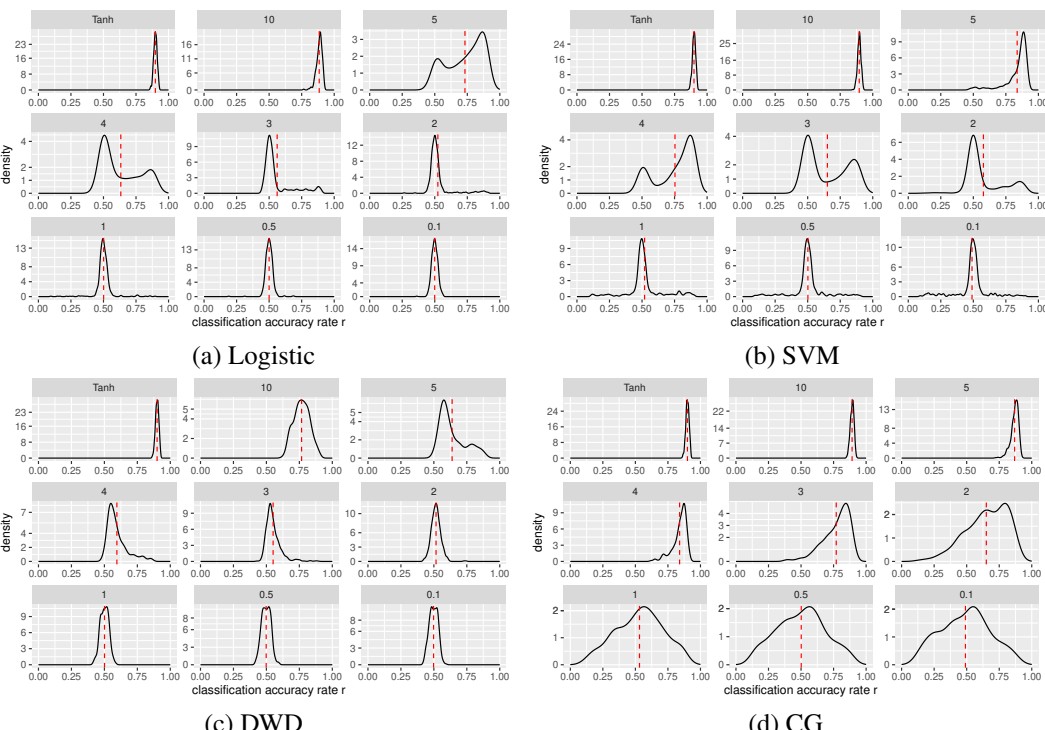

Figure 4: The empirical distribution $p_\varepsilon(r)$ of classifiers trained with $N = 3000$ clients and perturbed by Tanh transformation, where $\epsilon = 0.1, 0.5, 1, \ldots, 5, 10$, and the red dashed line represents the mean accuracy rate.

that classifiers with $d = 4, 5, 6$ exhibit comparable results.Therefore, we employ $d = 4$ cubic B-spline with equidistant knots to introduce as little noise as possible. We consider eight distinct privacy budget levels, specifically, $\epsilon = 0.1, 0.5, 1, \ldots, 5, 10$. Figure 3 displays the misclassification rates of the four types of classifiers mentioned in Section 3.2, based on both un-perturbed rescaled coefficients (i.e., "Tanh", "MM") and perturbed rescaled coefficients across various levels of $\epsilon$.

In Figure 3, with decreasing $\varepsilon$, the misclassification rates of different types of classifiers tend to 50%. At the same time, there is a variation in the performance of classifiers under the influence of noise. Notably, the misclassification rate of CG classifier remains highly volatile at smaller $\varepsilon$ values, ranging between 10% and 90%, instead of converging around 50% as other methods do. This indicates that the performance can be enhanced through model reversal and model averaging. Additionally, it can be observed from Figure 3 that classifiers based on both Tanh and Min-Max transformations exhibit very similar performances. Therefore, subsequent results will only showcase those based on the Tanh transformation.

In Theorem 3, we measure the enhancement in classification accuracy that model reversal can contribute to a classifier, characterized by its classification accuracy rate distribution $p_\varepsilon(r)$. The investigation of distribution $p_\varepsilon(r)$ itself is beyond the scope of this paper. However, we illustrate the empirical distribution of various classifiers under our experiment settings in Figure 4.

Figure 4 illustrates that different classifiers exhibit varying degrees of sensitivity to noise. Among them, the DWD classifier is the most affected, followed by logistic and SVM classifiers. In contrast, the CG classifier is relatively less impacted by noise interference. As $\varepsilon$ decreases, which corresponds to increased noise, the classification accuracy distributions $p_\varepsilon(r)$ for logistic, SVM, and DWD classifiers gradually converge around $0.5$. However, the distribution for the CG classifier remains more dispersed, indicating greater potential for improvement through model reversal.

### A.3 SAMPLE SIZE BALANCING

In this section, we assess the performance of classifiers over varying values of the parameters $N, N_0, N_1, n_0, n_1, B$. We generate data based on the settings presented in Section 5. Specifically, we consider five distinct combinations of these parameter values, which are listed in Table 2. Figures 5 and 6 display the misclassification rates of model-averaged classifiers using the cutoff value $r_0 = 0.6$, with and without model reversal, respectively, for the different parameter combinations.

Table 2: Five different combinations of parameters $N, N_0, N_1, n_0, n_1, B$

| Case | $N$ | $N_0$ | $N_1$ | $n_0$ | $n_1$ | $B$ |
|------|-----|-------|-------|-------|-------|-----|
| 1 | 3000 | 500 | 2500 | 100 | 100 | 25 |
| 2 | 5500 | 500 | 5000 | 100 | 100 | 50 |
| 3 | 5500 | 500 | 5000 | 50 | 100 | 50 |
| 4 | 3000 | 500 | 2500 | 100 | 50 | 50 |
| 5 | 3000 | 500 | 2500 | 50 | 50 | 50 |

In Figure 5, the differences arising from various parameter combinations on different types of classifiers manifest across distinct intervals of $\varepsilon$. Overall, Case 2 performs slightly better than Case 1, indicating that increasing the number of weak classifiers $B$ and the sample size of the validation set $N_1$ can enhance the performance. The preference for a relatively smaller $n_0$ is evident as Case 3 outperforms Case 2, and Case 5 outperforms Case 4. Additionally, Case 2 is distinctly superior to Case 4, and Case 3 is markedly better than Case 5, indicating a preference for a larger $n_1$. Importantly, by comparing Case 1 and Case 4, we discern that, given $N_0$ and $N_1$, the allocation should favor a relatively smaller $B$ and a larger $n_1$. This result is consistent with expectations, as a larger $n_1$ aids in more accurately estimating the performance of individual weak classifiers, thereby facilitating a more accurate model average.

Compared to Figure 5, Figure 6 incorporates model reversal prior to model averaging. It can be observed that the results of the various parameter combinations in Figure 6 are consistent with those in Figure 5. Notably, the introduction of model reversal has significantly enhanced the performance

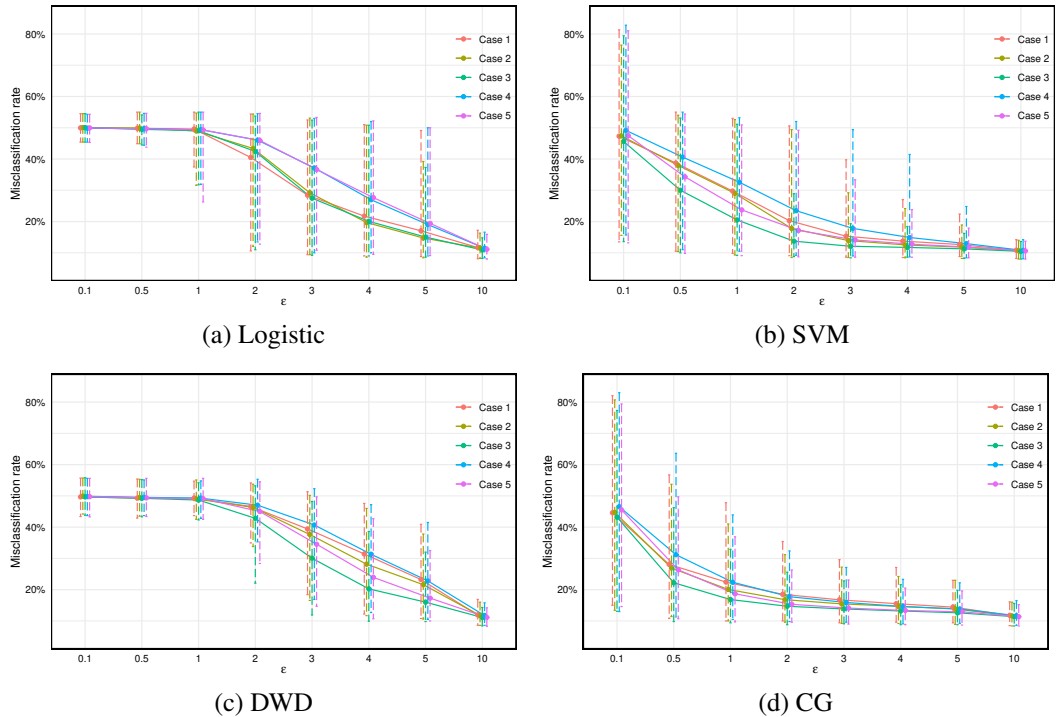

Figure 5: The misclassification rates with error bars of classifiers with model average under $\varepsilon$-LDP.

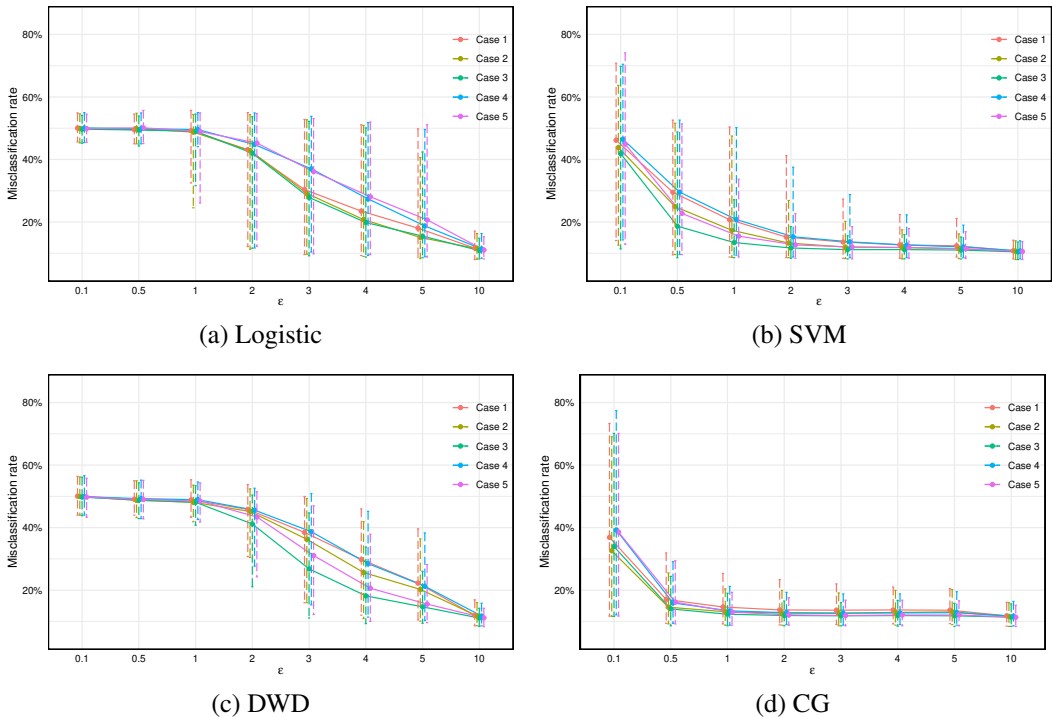

Figure 6: The misclassification rates with error bars of classifiers with model reversal and model average under $\varepsilon$-LDP.

of both SVM and CG classifiers, particularly under more stringent privacy protection levels, characterized by smaller intervals of $\varepsilon$.

## A.4  SAMPLE SIZE OF WEAK CLASSIFIER

To demonstrate how the number of clients used to train weak classifiers influences the efficacy of MRMA in improving classifier performance, we consider four distinct settings, which are listed in Table 3. Figure 7 shows how the misclassification rates of the classifier CG vary with different parameter settings.

We can see from the results of cases 5, 6, and 7 in Figure 7 that using more clients to train weak classifiers improves their performance when $\varepsilon$ is large, implying low noise levels. However, when $\varepsilon$ is small, which is our primary concern, increasing the sample size has little effect on the weak classifiers. In contrast, the classifiers based on MRMA consistently achieve significant improvements under different cases. Moreover, when we compare cases 7 and 8 in Figure 7, where the training and validation datasets have different proportions, we find that allocating more data for validation, i.e., for MRMA, leads to better results than using it to enhance weak classifiers.

Table 3: Four different combinations of parameters $N, N_0, N_1, n_0, n_1, B$

| Case | $N$ | $N_0$ | $N_1$ | $n_0$ | $n_1$ | $B$ |
|------|------|------|------|------|------|------|
| 5 | 3000 | 500 | 2500 | 50 | 50 | 50 |
| 6 | 5000 | 2500 | 2500 | 250 | 50 | 50 |
| 7 | 7500 | 5000 | 2500 | 500 | 50 | 50 |
| 8 | 7500 | 2500 | 5000 | 250 | 100 | 50 |

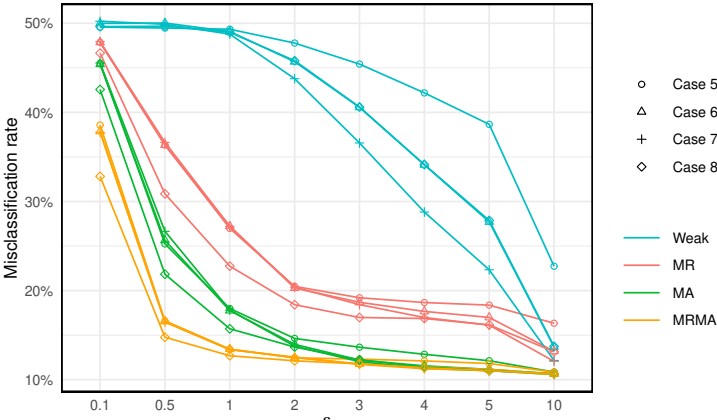

Figure 7: The misclassification rates of classifier CG with a single server under $\varepsilon$-LDP.

## A.5  VISUALIZATION OF FUNCTIONAL OBSERVATIONS

In this section, we visualize 200 randomly selected functional observations from groups "sh" and "iy" in the phonemes dataset. Figure 8 shows the raw functional observations and the curves reconstructed after finite basis projection (and transformation). The figure demonstrates that despite some loss of individual observational details due to finite basis projection and transformation, the fluctuating pattern of individual curves and the population-level group differences are still retained.

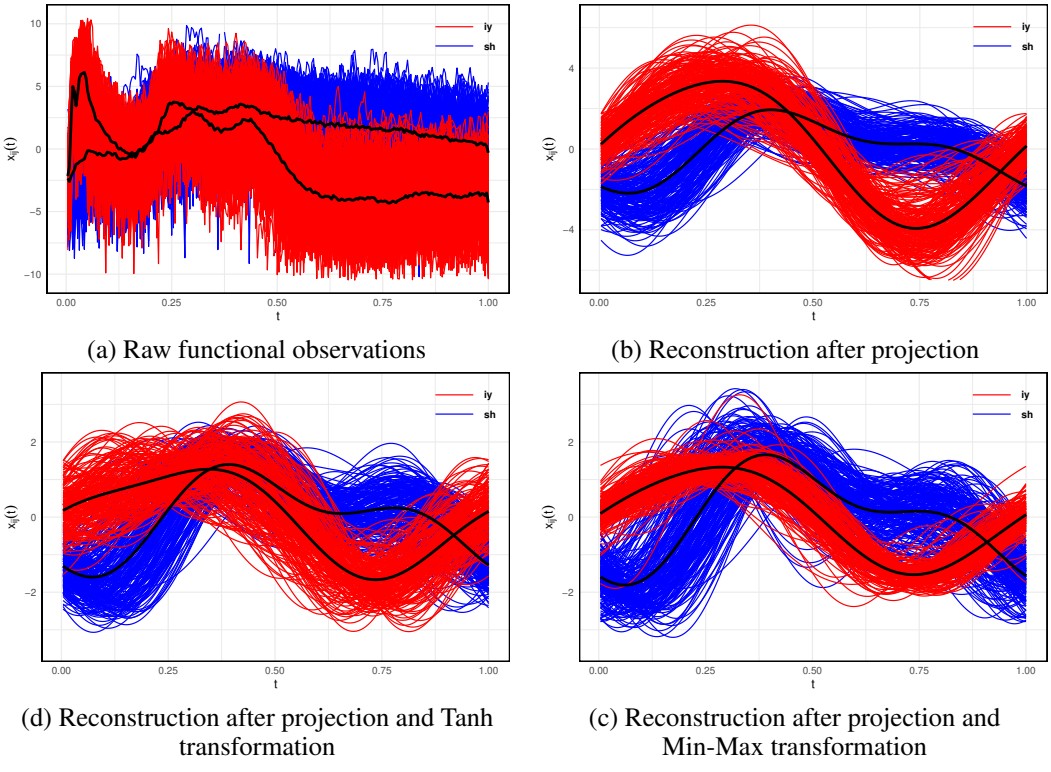

(a) Raw functional observations

(b) Reconstruction after projection

(d) Reconstruction after projection and Tanh transformation

(c) Reconstruction after projection and Min-Max transformation

Figure 8: Visualization of 200 randomly selected functional observations from groups "sh" and "iy" in the phonemes dataset. The two black lines in each figure represent the mean functions of the two groups of curves.

### A.6 MULTI-SERVER WITH FEDERATED LEARNING

In this section, we provide the details of building functional classifier under LDP with MRMA in heterogeneous multi-server settings in Algorithm 2. And we showcase the improvements in classification accuracy achieved through federated learning for individual servers. We consider three groups of servers, each characterized by a distinct slope function, to introduce heterogeneity among the servers. Of a total of $K = 25$ servers:

- Group 1: for $k = 1, \ldots, 10$, $\beta_k(t) = \sum_{j=1}^{50} \gamma_j(-1)^{j+1} j^{-2} \phi_j(t)$ with $\gamma_j \overset{i.i.d.}{\sim} U(-8, -2)$.
- Group 2: for $k = 11, \ldots, 15$, $\beta_k(t) \sim \text{GP}(0, K(s, t))$ with $K(s, t) = \exp(-15|s - t|)$.
- Group 3: for $k = 16, \ldots, 25$, $\beta_k(t) = \sum_{j=1}^{50} \gamma_j(-1)^{j+1} j^{-2} \phi_j(t)$ with $\gamma_j \overset{i.i.d.}{\sim} U(2, 8)$.

Here, $U(a, b)$ denotes a uniform distribution over the interval $(a, b)$, and $\text{GP}(0, K(s, t))$ represents a Gaussian process with zero mean and kernel $K(s, t)$. It is essential to highlight that the slope functions of servers within the same group are not identical. Moreover, the directions of the slope functions in groups 1 and 3 are opposite, serving as a test to assess the potential negative transfer impact during federated learning. In group 2, the slope function is randomly generated, resulting in an approximate 50% misclassification rate for classifiers built on servers in this group. This design is purposefully implemented to gauge its potential disruption to the federated learning process.

For each server, we generate $N = 3000$ clients for both training and validation, with an additional 500 clients designated for testing. Each server employs algorithms with the parameters $N_0 = 500$, $N_1 = 2500$, $n_0 = 50$, and $B = 50$. Initially, in the single-server context, each server independently runs Algorithm 1, setting the parameters $\varepsilon_v = \varepsilon$ and $r_0 = 0.7$. Subsequently, to deploy federated learning, servers execute Algorithm 2. Parameters for this phase are designated as $\varepsilon_v = \varepsilon_v^* = \varepsilon/2$. And we set $r_0 = 0.7$ to ensure the existence of weak classifiers satisfying this criterion, along with

---

**Algorithm 2** Functional Classification under LDP with MRMA and Federated Learning

1: **procedure** MULTI-SERVER
2:     **for** each server **do**
3:         $f_k^*(x) \leftarrow \textbf{Server}(\Phi, N_0, N_1, n_0, B, \varepsilon, \varepsilon_v = \varepsilon/2, r_0)$.
4:     **end for**
5:     **for** each server **do**
6:         Spilt clients in the validation set into $K$ subsets, $\mathcal{D}_{\text{valid}}^{(k)}, k = 1, \ldots, K$.
7:         **for** $k = 1, \ldots, K$ **do**
8:             **for** each client in the validation set $\mathcal{D}_{\text{valid}}^{(k)}$ **do**
9:                 $r_k' \leftarrow \textbf{ClientValid}(f_k^*, \varepsilon_v^* = \varepsilon/2)$
10:             **end for**
11:             Estimate $\tilde{r}^{(k)}$ by Equation 4.
12:             **if** $\tilde{r}^{(k)} < 50\%$ **then**
13:                 $f_k^* \leftarrow -f_k^*$                               ▷ Model Reversal
14:             **end if**
15:         **end for**
        Estimate $\boldsymbol{w} = (w_1, \ldots, w_K)^\top$ by Equation 5 with cutoff value $r_0^*$.
        **return** The final estimated classifier $f_k^\dagger$ by Equation 6.         ▷ Model Average
16:     **end for**
17: **end procedure**

---

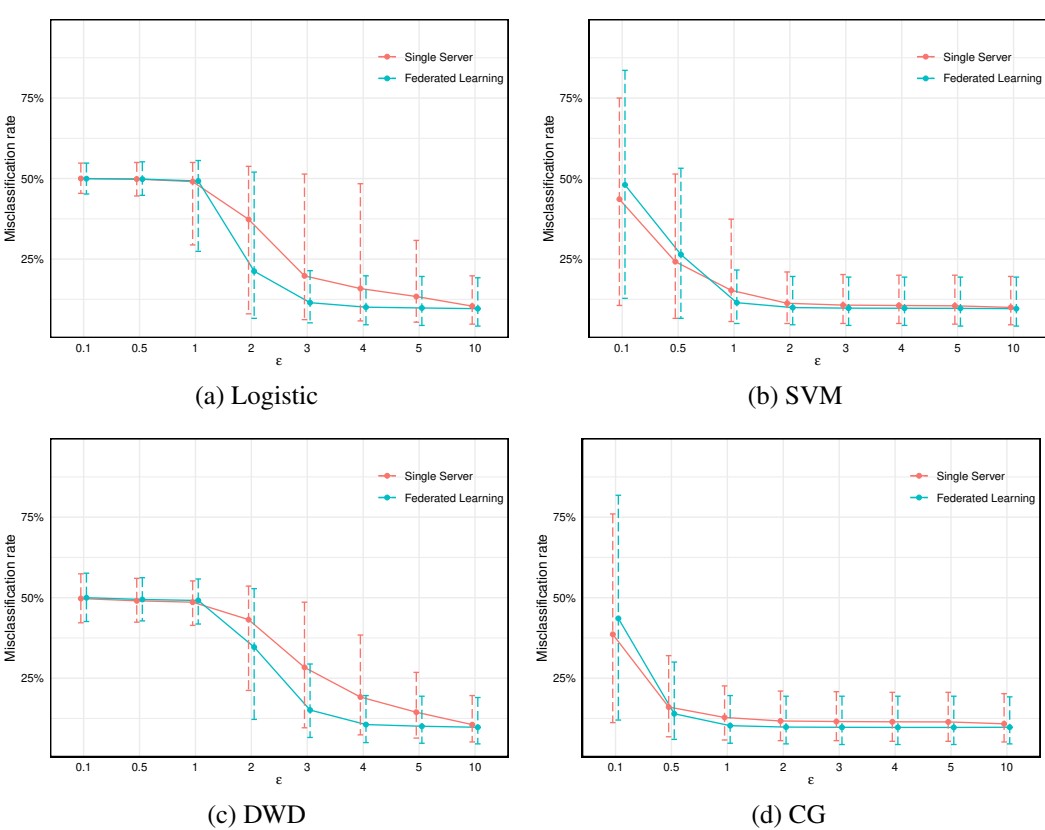

(a) Logistic                             (b) SVM

(c) DWD                               (d) CG

Figure 9: The misclassification rates of four types of classifiers with multi-server under $\varepsilon$-LDP.

$r_0^* = 0.8$ to counteract the potential negative transfer effect. As expected, servers in group 2 exhibit misclassification rates around 50%. The average misclassification rates for classifiers of servers in groups 1 and 3 under these two scenarios are illustrated in Figure 9. The results show that federated learning significantly improves the performance of both logistic and DWD classifiers, even with

server heterogeneity. While SVM and CG classifiers already perform well within a single server setting, federated learning shows comparable or slightly better performance.

# B  THEORETICAL ANALYSIS

## B.1  FUNCTIONAL PROJECTION AND INFORMATION LOSS

In this section, we provide further discussion on the projection-based functional classification, and measure the information loss induced by projection.

We regard functional observations $x(t)$ as random elements of the separable Hilbert space $L^2(\mathcal{I})$ of square-integrable functions on a compact domain $\mathcal{I}$ equipped with inner product $\langle f, g \rangle = \int_{\mathcal{I}} f(t)g(t)\mathrm{d}t$ and norm $\|f\| = \langle f, f \rangle^{1/2}$. We consider classification of a Gaussian random function, $X(t)$, into one of two classes of Gaussian random functions with mean function $\boldsymbol{\mu}_0$ and $\boldsymbol{\mu}_1$, respectively. Both classes have covariance operator $\mathcal{K}$ defined as the integral operator

$$(\mathcal{K}f)(\cdot) = \int_{\mathcal{I}} \rho(\cdot, t)f(t)\mathrm{d}t$$

with kernel $\rho(s, t) = \mathrm{cov}\{X(s), X(t)\}$.

Under the assumption that $\boldsymbol{\mu}_0, \boldsymbol{\mu}_1$ and $\mathcal{K}$ are known, Delaigle & Hall (2012) and Kraus & Stefanucci (2019) consider the class of centroid classifiers that are based on one-dimensional projections of the form $\langle X, \psi \rangle$, where $\psi$ is a function in $L^2(\mathcal{I})$. In this section, we study the classifier based on multi-dimensional projections.

**Lemma 1.** *If $\mathcal{K}^{-1}(\boldsymbol{\mu}_1 - \boldsymbol{\mu}_0)$ is in the span of projection function $\Phi_d = (\phi_1, \ldots, \phi_d)^\top$, then $\Phi_d$ is optimal in the sense that the classifier $\mathcal{C}_{\Phi_d}$ can achieve the lowest possible misclassification rate*

$$1 - G\left(\frac{\langle \boldsymbol{\mu}, \mathcal{K}^{-1}\boldsymbol{\mu} \rangle^{1/2}}{2}\right)$$

*among all possible classifiers, where $G$ is the standard normal cumulative distribution function.*

Lemma 1 clearly demonstrates that the classifier based on finite projection functions can be optimal. Specifically, in the case where $d = 1$, the function $\phi_1 = \mathcal{K}^{-1}(\boldsymbol{\mu}_1 - \boldsymbol{\mu}_0)$ serves as an optimal projection function, efficiently reducing functional observations to a scalar.

**Proof of Lemma 1.** For any function $x(t)$ and a set of basis functions denoted as $\Phi_d = (\phi_1, \ldots, \phi_d)^\top$, we define $\Phi_d x = (\langle \phi_1, x \rangle, \ldots, \langle \phi_d, x \rangle)^\top \in \mathbb{R}^d$, and $\Phi_d \mathcal{K} \Phi_d^\top$ as a $d \times d$ matrix, where the $(k_1, k_2)$th element is $\langle \phi_{k_1}, \mathcal{K}\phi_{k_2} \rangle$. With a slight abuse of notation, we represent $\Phi_d$ as $\Phi$ in the following text. For any given $d$-dimension vector $\boldsymbol{\theta}$, record $\psi = \Phi^\top \boldsymbol{\theta} = \sum_{k=1}^d \theta_k \phi_k$. Let the mean function of class $i$ be represented as $\boldsymbol{\mu}_i$ for $i = 0, 1$. The optimal classifier based on $\langle X, \psi \rangle$ assigns $X$ to the class $\mathcal{C}_{\Phi, \boldsymbol{\theta}}(X)$ given by

$$\mathcal{C}_{\Phi, \boldsymbol{\theta}}(X) = I\left(\langle X - \boldsymbol{\mu}_0, \psi \rangle^2 - \langle X - \boldsymbol{\mu}_1, \psi \rangle^2 > 0\right) = I\left(T_{\Phi, \boldsymbol{\theta}}(X) > 0\right),$$

where $T_{\Phi, \boldsymbol{\theta}}(X) = \langle X - \bar{\boldsymbol{\mu}}, \psi \rangle \langle \boldsymbol{\mu}, \psi \rangle$ with $\bar{\boldsymbol{\mu}} = (\boldsymbol{\mu}_0 + \boldsymbol{\mu}_1)/2$ and $\boldsymbol{\mu} = \boldsymbol{\mu}_1 - \boldsymbol{\mu}_0$. The misclassification rate of this classifier is

$$D(\Phi, \boldsymbol{\theta}) = P_0\left(\mathcal{C}_{\Phi, \boldsymbol{\theta}}(X) = 1\right)/2 + P_1\left(\mathcal{C}_{\Phi, \boldsymbol{\theta}}(X) = 0\right)/2 = 1 - G\left(\frac{|\langle \boldsymbol{\mu}, \Phi^\top \boldsymbol{\theta} \rangle|}{2\langle \Phi^\top \boldsymbol{\theta}, \mathcal{K}\Phi^\top \boldsymbol{\theta} \rangle^{1/2}}\right), \quad \text{(B.1)}$$

where $P_i$ is the distribution of curves in class $i$, $i = 0, 1$.

By minimizing Equation B.1, we can find the optimal selection of the projection functions $\Phi$ and its corresponding $\boldsymbol{\theta}$. Firstly, for a given $\Phi$, by the Cauchy–Schwarz inequality, if $\|(\Phi\mathcal{K}\Phi^\top)^{-1/2}\Phi\boldsymbol{\mu}\| < \infty$, we have

$$\frac{\langle \boldsymbol{\mu}, \Phi^\top \boldsymbol{\theta} \rangle^2}{\langle \Phi^\top \boldsymbol{\theta}, \mathcal{K}\Phi^\top \boldsymbol{\theta} \rangle} \leq \|(\Phi\mathcal{K}\Phi^\top)^{-1/2}\Phi\boldsymbol{\mu}\|^2.$$

If $\|(\Phi\mathcal{K}\Phi^\top)^{-1}\Phi\boldsymbol{\mu}\| < \infty$, the equality is achieved for $\boldsymbol{\theta} = (\Phi\mathcal{K}\Phi^\top)^{-1}\Phi\boldsymbol{\mu} := \boldsymbol{\theta}_0$. And the corresponding misclassification rate is

$$D(\Phi, \boldsymbol{\theta}_0) = 1 - G\left(\frac{\|(\Phi\mathcal{K}\Phi^\top)^{-1/2}\Phi\boldsymbol{\mu}\|}{2}\right). \quad \text{(B.2)}$$

This implies that for any given $\Phi$ with $d > 1$, $\psi_0 =: \Phi^\top \boldsymbol{\theta}_0$ is the most efficient projection function among all the linear combinations of $\Phi$. Similarly, by minimizing $D(\Phi, \boldsymbol{\theta}_0)$, we find that the optimal projection functions $\Phi^*$ is the $\Phi$ that satisfies

$$\Phi^\top \boldsymbol{\theta}_0 = \mathcal{K}^{-1} \boldsymbol{\mu}, \tag{B.3}$$

and the corresponding misclassification rate is

$$D(\Phi^*, \boldsymbol{\theta}_0) = 1 - G\left(\frac{\langle \boldsymbol{\mu}, \mathcal{K}^{-1} \boldsymbol{\mu} \rangle^{1/2}}{2}\right), \tag{B.4}$$

which is the lowest possible misclassification rate for this problem among all possible classifiers (Berrendero et al., 2018). $\qquad\square$

In practice, selecting a projection function $\Phi$ that satisfies Equation B.3 is difficult since both $\boldsymbol{\mu}$ and $\mathcal{K}$ are unknown. For a general projection function $\Phi$, the misclassification rate of the classifier based $\Phi$ is given in Equation B.2. We can measure the information loss caused by $\Phi$ relative to the optimal one, $\Phi^*$, by

$$\langle \boldsymbol{\mu}, \mathcal{K}^{-1} \boldsymbol{\mu} \rangle - \|(\Phi \mathcal{K} \Phi^\top)^{-1/2} \Phi \boldsymbol{\mu}\|^2$$
$$= \boldsymbol{\mu}^\top \mathcal{K}^{-1/2} (I - P_{\mathcal{K}^{1/2} \Phi^\top}) \mathcal{K}^{-1/2} \boldsymbol{\mu} =: \|\boldsymbol{\delta}\|^2 \geq 0,$$

where $P_{\mathcal{K}^{1/2} \Phi^\top} = \mathcal{K}^{1/2} \Phi^\top (\Phi \mathcal{K} \Phi^\top)^{-1} \Phi \mathcal{K}^{1/2}$. The loss $\|\boldsymbol{\delta}\|^2$ is exactly the sum of squares error of the regression model

$$\mathcal{K}^{-1/2} \boldsymbol{\mu} = \mathcal{K}^{1/2} \Phi^\top \boldsymbol{\theta},$$

when the coefficient $\boldsymbol{\theta} = \boldsymbol{\theta}_0$.

## B.2 PROOF OF THEOREMS

**Proof of Theorem 1**. Let $u = (x(t), y)$ represent the possible observation of a client, and $v = (\boldsymbol{z}', y')$ represent the possible report of a client. Let $f_{v|u}(v|u)$ be the conditional density of $v$ given $u$. Then for any possible output $v$ of $\mathcal{M}_1(u)$, by the sequential composition theorem (McSherry & Talwar, 2007), we have

$$\frac{f_{v|u}(v|u_1)}{f_{v|u}(v|u_2)} = \frac{P(y'|y_1)}{P(y'|y_2)} \prod_{k=1}^d \frac{f(z_k' - z_{1,k}^*)}{f(z_k' - z_{2,k}^*)}$$

$$\leq \max(1, \frac{q}{1-q}, \frac{1-q}{q}) \prod_{k=1}^d \exp(\frac{\varepsilon_1}{d\Delta}(|z_k' - z_{1,k}^*| - |z_k' - z_{2,k}^*|))$$

$$\leq e^{\varepsilon_2} \prod_{k=1}^d \exp(\frac{\varepsilon_1}{d}) = e^{\varepsilon_1 + \varepsilon_2} = e^\varepsilon,$$

where $q = e^{\varepsilon_2}/(1 + e^{\varepsilon_2})$. Thus $\mathcal{M}_1$ satisfies $\varepsilon$-local differential privacy.

**Proof of Theorem 2**. Let $u = (x(t), y)$ represent the possible observation of a client, and $w = r'$ represent the possible report of a client. Let $f_{w|u}(w|u)$ be the conditional density of $w$ given $u$. Then for any $w \in \{0, 1\}$, we have

$$\frac{f_{w|u}(w|u_1)}{f_{w|u}(w|u_2)} = \frac{P(r'|r_1)}{P(r'|r_2)} \leq e^{\varepsilon_v}.$$

Thus $\mathcal{M}_2$ satisfies $\varepsilon_v$-local differential privacy. Note that $r^{(b)}$ is the classification accuracy of the classifier $f^{(b)}$, then

$$\mathbb{E}(\hat{r}^{(b)}) = qr^{(b)} + (1-q)(1 - r^{(b)}),$$

where $q = e^{\varepsilon_v}/(1 + e^{\varepsilon_v})$. Thus we have $\mathbb{E}(\tilde{r}^{(b)}) = r^{(b)}$, and $\text{Var}(\tilde{r}^{(b)}) = \text{Var}(\hat{r}^{(b)})/(2q - 1)^2 \leq ((e^{\varepsilon_v} + 1)/(e^{\varepsilon_v} - 1))^2/(4n_1^{(b)})$. This concludes the proof.

**Proof of Theorem 3**. For any $r \in [0, 1]$, we have $r_\delta = \max(1 - 2r, 0)$, thus

$$\mathbb{E}(r_\delta) = \int_0^1 p_\varepsilon(r) \max(1 - 2r, 0) dr = \int_0^{1/2} p_\varepsilon(r)(1 - 2r) dr.$$

