# OpenReview forum: "Functional Classification Under Local Differential Privacy with Model Reversal and Model Average"
_ICLR.cc/2024/Conference — Submitted to ICLR 2024_

### Official Review · Reviewer_WLyT · 2023-10-26

**Soundness:** 2 fair
**Presentation:** 3 good
**Contribution:** 1 poor
**Rating:** 3
**Confidence:** 3

**Summary:**

The authors have provided a method for adding noise to functional classifiers to satisfy Local Differential Privacy (LDP). In the context of federated learning, the authors introduce the idea of using more clients to evaluate the performance of weak classifiers and propose the concept of model reversal, which involves inverting the parameters of models with poor accuracy. Additionally, the authors propose a federated learning approach for the heterogeneous multi-server setting under LDP.

**Strengths:**

1.The paper introduces a novel approach to functional data classification under Local Differential Privacy (LDP), a topic that has not been extensively explored in the existing literature.
2.The introduction of model reversal is particularly innovative, as it provides a unique solution to enhance the performance of weak classifiers by inverting their parameters when their accuracy is below a certain threshold.

**Weaknesses:**

1.Differential Privacy (DP) has already proven to be sufficiently effective in protecting individual privacy, for instance, in defending against membership inference attacks. Moreover, there are numerous existing methods within DP for adding noise to functional input data, where the idea of projecting onto a set of finite bases is also quite common. From this perspective, the innovation of projection might not be high enough, and the necessity of employing Local Differential Privacy (LDP) remains to be further examined.At the same time, the performance degradation caused by residuals has not been analyzed theoretically. The expressive power of a finite basis is quite limited, and for some functions, their residuals could be fatal.

2.Regarding the ideas of model average and model reversal proposed by the authors, compared to traditional ensemble techniques, model average utilizes more clients in the evaluation process, aiming to select better weak classifiers. For each poor classifier, a model reversal approach is employed for improvement. However, since only a small number of clients are used to train the parameters, the classifiers might generally perform poorly; model reversal can enhance the performance of the learners, but it may not necessarily improve them to a satisfactory level. Theorem three provides an expected conclusion, but it does not directly lead to a high-probability bound; the theoretical analysis of model reversal is still lacking.

**Questions:**

1..Are there specific scenarios or use cases where LDP provides a clear advantage over DP in functional data classification?

2.The idea of projecting data onto a finite basis is quite common in the realm of DP. Could you clarify how your approach innovates or differs significantly from existing methods?

3.The theoretical analysis of model reversal seems to be lacking. Could you provide more details or elaborate on the theoretical foundations of this technique?

---

> ### Author Response · Authors · 2023-11-20
> **Response to Reviewer WLyT (part 1/4)**
>
> We are pleased that the reviewer found our model reversal approach particularly innovative. We thank the reviewer for the valuable comments and suggestions for improvement, and we have incorporated relevant discussions and modifications into our paper. Below, we provide detailed responses to the reviewer's questions, aiming to encourage a reconsideration of our paper's evaluation.
>
> ### Weakness 1 (1/2):
>
> > Differential Privacy (DP) has already proven to be sufficiently effective in protecting individual privacy, for instance, in defending against membership inference attacks. Moreover, there are numerous existing methods within DP for adding noise to functional input data, where the idea of projecting onto a set of finite bases is also quite common. From this perspective, the innovation of projection might not be high enough, and the necessity of employing Local Differential Privacy (LDP) remains to be further examined.
>
> We thank the reviewer for the insightful comments. In response to these concerns, we provide responses from the following two perspectives.
>
> 1. **A comparison of DP and LDP**: Indeed, **Differential Privacy (DP)** is a foremost framework for privacy-preserving statistical analyses. It offers a rigorous and interpretable definition of data privacy, limiting the amount of information that attackers can infer from publicly released database queries. In most scenarios, the privacy protection required can be adequately addressed with DP. **Local Differential Privacy (LDP)**, on the other hand, is designed for more extreme cases, such as when the data collector is untrustworthy or malicious, or when privacy must be safeguarded right from the data collection stage, particularly in instances involving highly sensitive or personal data. **For example**, LDP is crucial in situations like smart home devices collecting user behavior data, mobile keyboard apps processing sensitive text inputs, and health monitoring wearables tracking personal health metrics.
>
> 2. **Functional data analysis (FDA) under LDP vs DP**: As the reviewer pointed out, there are existing works about projecting data onto a finite basis within DP. In the revised manuscript's **Section 2**, we've summarized existing DP methods for adding noise to functional data. However, FDA under LDP and DP faces different challenges. **First of all**, under DP, the server can access all the raw data and achieve privacy protection by utilizing knowledge of the covariance function and sensitivity of functional objects. In contrast, LDP requires each functional object to be privatized before senting to the server. **Furthermore**, under LDP, the emphasis is on the performance of models trained on the noised functional observations from clients, rather than on the recovery of functional data.

---

> ### Author Response · Authors · 2023-11-20
> **Response to Reviewer WLyT (part 2/4)**
>
> ### Weakness 1 (2/2):
>
> > At the same time, the performance degradation caused by residuals has not been analyzed theoretically. The expressive power of a finite basis is quite limited, and for some functions, their residuals could be fatal.
>
> Thanks for pointing out the problem about expressive power of a finite basis. In the revised manuscript, we offered a detailed discussion on finite basis projection from various perspectives in **Section 3.1**, present some **theoretical analysis in Appendix B.1**, and provide a more intuitive demonstration using **real data in Appendix A.5**. We explained the acceptability of finite basis projection under LDP from the following perspectives:
>
> 1. **Functional data fitting**: According to [Eubank, 1999], when fitting functional observations with common basis functions, the number of basis needed for optimal estimation risk increases slowly with the number of observed time points $T$. Taking cubic B-Spline (with order $m=4$) as an example, the estimation risk decays at the optimal rate of $T^{-2m/(2m+1)}$ as the number of inner knots $K$ grows like $T^{1/(2m+1)}$. With the number of basis $d=m+K$, finite B-Spline basis can effectively fit the data;
>
> 2. **Performance of projection-based classifiers**: **Lemma 1 in Appendix B.1** demonstrates that under certain conditions, classifiers based on projection with finite basis can achieve the lowest possible misclassification rate among all possible classifiers;
>
> 3. **Functional projection under LDP**: Projection-based methods enable the server to efficiently capture data patterns of interest, such as using Fourier basis for specific frequency bands or B-Spline for certain intervals. Employing finite basis helps capture population-level patterns while preventing overfitting to individual differences and enhances communication efficiency;
>
> 4. **Real data**: **Figure 8 in Appendix A.5** visualizes functional observations from a phonemes dataset analyzed in **Section 6**. It also shows the curves reconstructed after finite basis projection and transformation. The projection (and transformation) erases individual detail information but retains the fluctuating pattern of individual curves, and the population-level group differences is also preserved.
>
> [Eubank, 1999] Randall L Eubank. Nonparametric regression and spline smoothing. CRC press, 1999.

---

> ### Author Response · Authors · 2023-11-20
> **Response to Reviewer WLyT (part 3/4)**
>
> ### Weakness 2:
>
> > Regarding the ideas of model average and model reversal proposed by the authors, compared to traditional ensemble techniques, model average utilizes more clients in the evaluation process, aiming to select better weak classifiers. For each poor classifier, a model reversal approach is employed for improvement. However, since only a small number of clients are used to train the parameters, the classifiers might generally perform poorly; model reversal can enhance the performance of the learners, but it may not necessarily improve them to a satisfactory level. Theorem three provides an expected conclusion, but it does not directly lead to a high-probability bound; the theoretical analysis of model reversal is still lacking.
>
> Thanks for pointing out the potential issues within Model Reversal (MR). We have addressed these in our revised manuscript by adding experimental results of classifiers based on MR, including experiments with weak classifiers trained using varying sample sizes, and by revising the content of Theorem 3.
>
> **Experimental Performance of MR**
>
> - From **Figure 1 in Section 5**, we observed that MR significantly improves the performance of all types of weak classifiers. And compared to Model Average (MA), the MRMA-based classifiers further substantially enhances the performance of SVM and CG classifiers;
> - In **Appendix A.4**, experimental results show that the number of clients used to train weak classifiers does not affect the efficacy of MR or MRMA in improving classifier performance. Additionally, when more client data is available, allocating more data for validation (i.e., for MR and MA) yields better results than using it solely to strengthen weak classifiers.
>
> **Theoretical Analysis of MR**
>
> - Given a classifier's accuracy rate distribution under $\varepsilon$-LDP as $p_\varepsilon(r)$ with $r\in[0,1]$, **Theorem 3** measures the potential improvement in classification accuracy through MR;
>
> - Although a detailed theoretical analysis of the distribution $p_\varepsilon(r)$ for various classifiers is beyond this paper's scope, it's evident from **Figure 1 in Section 5** and **Figure 3 in Appendix A.2** that different weak classifiers have varying sensitivities to noise and exhibit different distributions $p_\varepsilon(r)$. This variation explains why MR offers more improvement for SVM and CG classifiers than Logistic and DWD classifiers in experiments, especially at lower values of $\varepsilon$.

---

> ### Author Response · Authors · 2023-11-20
> **Response to Reviewer WLyT (part 4/4)**
>
> ### Q1) Are there specific scenarios or use cases where LDP provides a clear advantage over DP in functional data classification?
>
> We apologize for not clearly distinguishing between LDP and DP in the original manuscript. As mentioned in the revised manuscript's Introduction and in response to **Weakness 1 (1/2)**, DP is sufficient for privacy protection in most scenarios. LDP is designed for more extreme cases, such as when dealing with untrustworthy or malicious data collectors, especially in situations involving highly sensitive or personal data. Here are two specific examples concerning remote health monitoring and fitness data where it is necessary to add noise at the individual data level before centralization:
>
> 1. **Remote Health Monitoring Scenario** Consider a scenario where a remote health monitoring system is implemented by a healthcare provider. In this scenario, continuous health data, such as heart rate, blood pressure, and sleep patterns, are collected from patients through wearable devices. This sensitive data, once acquired by an untrusted server, could potentially be used to infer medical conditions or health status categories of the patients. Such inferences could then be utilized for various purposes, ranging from personalized healthcare and treatment plans to potentially more intrusive uses like health insurance premium adjustments or targeted advertising based on perceived health risks.
>
> 2. **Fitness Data Collection Scenario** In another scenario, a fitness app collects detailed data on user activities, including step count, exercise routines, and physiological responses to workouts, from wearable fitness trackers. If this data falls into the hands of an untrusted server, it could be used to deduce categories such as fitness levels, lifestyle habits, or even underlying health issues of the users. Consequently, this information might be exploited for purposes like customizing fitness programs, but it also raises concerns about privacy breaches, where users’ fitness data could be used for unsolicited health product marketing or influence health insurance policies without the users' explicit consent.
>
> ### Q2) The idea of projecting data onto a finite basis is quite common in the realm of DP. Could you clarify how your approach innovates or differs significantly from existing methods?
>
> We apologize for not making everything clear. The idea of projecting data onto a finite basis is indeed a commonly employed method. However, as we discussed in **Section 2** of the revised manuscript and in our response to **Weakness 1 (1/2)**, functional data analysis under DP or LDP faces distinctly different challenges. In the context of LDP, privacy learning with functional data, through finite basis projection, effectively reduces dimensionality to lessen noise interference, while simultaneously ensuring less impact on the efficacy of the learned models. Our paper is the first to consider privacy learning with functional data under LDP. We also propose novel techniques, Model Reversal and Model Average, tailored for LDP, to enhance classifier performance.
>
> For a detailed overview of the novelty of our work, we refer to **General response** for a discussion.
>
> ### Q3) The theoretical analysis of model reversal seems to be lacking. Could you provide more details or elaborate on the theoretical foundations of this technique?
>
> We thank the reviewer for the kind advice. In the revised version of our paper, we have updated the content of **Theorem 3 in Section 3.3** and provided some discussion on it. And in our response to **Weakness 2**, we have discussed in detail the theoretical analysis of Model Reversal. We hope this adequately addresses the reviewer's concerns.
>
>
>
> Once again, we appreciate the reviewer's valuable time. We look forward to the opportunity to engage in more detailed discussions to dispel any concerns.

---

> ### Author Response · Authors · 2023-11-22
> **Seeking Feedback on Our Responses and Revised Submission**
>
> Dear Reviewer WLyT,
>
> Thank you once again for your valuable insights and the time you have dedicated to reviewing our work. Your feedback has been instrumental in enhancing our revised manuscript, and we are keen to hear any additional comments or suggestions you might have.
>
> With the discussion deadline nearing, we would be grateful for your reply to ensure that all your concerns have been adequately addressed in our latest revision. Your guidance is invaluable to us, and we eagerly await your response at your earliest convenience.
>
> Best regards!

---

### Official Review · Reviewer_D9bv · 2023-10-30

**Soundness:** 3 good
**Presentation:** 3 good
**Contribution:** 2 fair
**Rating:** 5
**Confidence:** 3

**Summary:**

Functional data is infinite-dimensional data which can be approximated by a linear combination of basis functions; this paper explores how to classify functional data under the constraint of local differential privacy (LDP). The authors demonstrate how to construct “weak” functional data classifiers under LDP, and then demonstrate how to boost the weak classifiers’ performance using model averaging (which combines the weak classifiers) and the novel technique of “model reversal” (which flips the signs of a weak classifier’s coefficients). The authors combine these techniques into algorithms for both single-server and multi-server (i.e., federated learning) settings.

**Strengths:**

1. The problem setting as well as the proposed algorithms are quite novel. In particular, I think that the idea of model reversal is kind of fun and creative.

2. The experimental results also seem to show that using the proposed methods does improve classification accuracy.

**Weaknesses:**

1. The setting seems obscure and not very well-motivated. I don’t really understand the significance of functional data and where it’s found in real-world applications.

2. I’m not fully convinced by the empirical evaluation. The experimental results show that the proposed techniques improve classification performance, but at the same time the baselines don’t set a particularly high bar to beat. While I understand that a little-explored setting won’t have much previous work to compare to, it is still hard to judge the importance of the methods without them. And touching back on Weakness #1, the experiments are all conducted on a synthetic dataset and so it’s unclear how well the methods will generalize to real-world data.

**Questions:**

1. Would it be possible to give more details on the practicality of functional data? For example, are there realistic datasets that are considered to be “functional data” and how do the properties of functional data translate to the real world?

2. Is there a formal definition for a “weak classifier”?

3. The functional data classifiers are constructed under LDP, and then it seems to me that the LDP setting is kind of irrelevant to the rest of the paper (or at least, to the model average and model reversal techniques). Similarly for the functional data setting, after dimensionality reduction and re-scaling. Do MA and MR actually have anything to do specifically with LDP (or with functional data), or are they general techniques that could be used to improve the performance of any collection of weak classifiers?

---

> ### Author Response · Authors · 2023-11-20
> **Response to Reviewer D9bv (part 1/2)**
>
> We are delighted that reviewer found the idea of Model Reversal fun and creative. We thank the reviewer for the insightful and constructive comments. Below, we provide detailed responses to each comment and explain how we have revised our manuscript accordingly.
>
> ### W1) The setting seems obscure and not very well-motivated. I don’t really understand the significance of functional data and where it’s found in real-world applications.
>
> Thanks for pointing it out! We've realized our article lacked an introduction to functional data and its privacy aspects. In response, we have now included in **Section 1 (Introduction)** a brief overview of functional data and the importance of privacy preservation, along with relevant examples. Additionally, **Section 6 (Real Application)** has been updated to include a real data application to demonstrate these concepts in practice.
>
> - **Functional data analysis (FDA)** is a branch of statistics that deals with data that is densely observed over a continuum, such as time or space. This type of data is distinct from traditional multivariate data, which typically consists of a finite number of variables measured at a discrete set of points. FDA is widely used in various fields **such as** economics (for analyzing financial time series), environmental sciences (for studying climate change effects), health sciences (for analyzing continuous health monitoring data), and engineering (for signal processing). And privacy preservation in functional data is also crucial in domains **such as** health informatics, where hospital patient monitoring systems can reveal sensitive health information, and in behavioral science, where data from smart devices can provide deep insights into an individual's lifestyle and habits.
>
> ### W2) I’m not fully convinced by the empirical evaluation.
>
> > The experimental results show that the proposed techniques improve classification performance, but at the same time the baselines don’t set a particularly high bar to beat. While I understand that a little-explored setting won’t have much previous work to compare to, it is still hard to judge the importance of the methods without them. And touching back on Weakness #1, the experiments are all conducted on a synthetic dataset and so it’s unclear how well the methods will generalize to real-world data.
>
> We thank the reviewer for the insightful comments. In the revised version of our paper, we have supplemented it with comparative experiments and added a real-world application.
>
> - **Experimental Results**: In **Figure 1 of Section 5**, we supplemented results for classifiers based on classic aggregation methods, "Voting" and "Averaging". It’s evident that under LDP, these classic methods are ineffective. However, our proposed techniques, both Model Reversal (MR) and Model Average (MA), significantly improve the performance of all types of weak classifiers, even at lower $\varepsilon$ values, which corresponds to higher levels of privacy protection. MRMA, in particular, substantially further enhances the performance of SVM and CG classifiers. Similar results can also be found in **Figure 7 in Appendix A.4**. As the reviewer mentioned, since this is the first paper to study functional classification, our simulations lacked existing methods for comparison. If there are any additional suggestions for comparison, please let us know.
>
> - **Real Application**: In **Section 6 (Real Application)**, we employed our methods on a phonemes dataset derived from the TIMIT Speech Corpus. To ensure the robustness and reliability of our results, we conducted 500 random splits of the dataset into test, training, and validation sets. The performance of various methods in this real-world application was similar to that observed in the simulations, with classifiers based on MR and MA still showing good performance.

---

> ### Author Response · Authors · 2023-11-20
> **Response to Reviewer D9bv (part 2/2)**
>
> ### Q1) Would it be possible to give more details on the practicality of functional data?
>
> > For example, are there realistic datasets that are considered to be "functional data" and how do the properties of functional data translate to the real world?
>
> We thank the reviewer for the kind advice. In the revised version of our paper, we introduced functional data and provided examples of its practical applications in the first paragraph of **Introduction**. And in our response to **W1)**, we give functional data examples in various fields. We hope this sufficiently addresses the reviewer's concern.
>
> ### Q2) Is there a formal definition for a "weak classifier"?
>
> I apologize for the lack of clarity in our paper. We introduced the process of generating weak classifiers during the training phase in **Section 3.3** of our paper. And in the context of machine learning, a "weak classifier" generally refers to a simple model that performs only slightly better than a random guesser in classifying instances into classes. This concept is often discussed in ensemble learning methods, particularly in boosting algorithms.
>
> ### Q3) The functional data classifiers are constructed under LDP [...]
>
> > The functional data classifiers are constructed under LDP, and then it seems to me that the LDP setting is kind of irrelevant to the rest of the paper (or at least, to the model average and model reversal techniques). Similarly for the functional data setting, after dimensionality reduction and re-scaling. Do MA and MR actually have anything to do specifically with LDP (or with functional data), or are they general techniques that could be used to improve the performance of any collection of weak classifiers?
>
> We apologize for not making the relationship among LDP, functional data, and MRMA clear. In this paper, we proposed a new technique, **Model Reversal**, to enhance the performance of a single weak classifier under LDP. We also introduced a **Model Average** approach tailored for LDP to obtain a better collection of weak classifiers. And the data allocation ratio for training and validation sets, as well as the validation process used in our MRMA technique, are all specifically designed for LDP.
>
> Here, we provide a more detailed explanation of the relationship among LDP, functional data, and MRMA:
>
> - **LDP and MR (Model Reversal)**: MR is a new technique we developed to improve the performance of weak classifiers under LDP. Under LDP, it's common for weak classifiers to have misclassification rates greater than $50\%$ due to high noise in the data, and using MR can lead to significant improvements. In **Theorem 3 of Section 3.3**, we measured the enhancement that model reversal can bring to the classifier;
>
> - **LDP and MA (Model Average)**: MA in our paper is designed specifically for LDP challenges. Specifically, we proposed a method to evaluate the performance of weak classifiers under LDP in **Theorem 2 of Section 3.3**. And based on the evaluation, we assigned more suitable weights to these classifiers during the collection process;
>
> - **Functional data and MRMA**: Our MRMA algorithm is applicable to functional data as well as other data types. After obtaining weak classifiers trained on various data types under LDP, it can be used to construct a better-performing classifier.
>
>
>
> Once again, we are grateful for the time and effort the reviewer has invested in evaluating our work. And we welcome any additional dialogue to clarify any aspects of our research.

---

> > ### Comment · Reviewer_D9bv · 2023-11-21
> >
> > Thanks very much for the detailed clarifications. The revised paper is looking good!
> >
> > Most of my concerns have been addressed, but I do share Reviewer sTQT’s sentiment that the techniques proposed in the paper aren’t tailored specifically to the problem setting (of functional data and LDP). For example, I agree that model reversal seems advantageous in the LDP setting (because the large amount of noise required by LDP could cause weak classifiers to have misclassification rates greater than 50%). But it’s still not clear to me how or if the functional data setting is relevant to the proposed algorithms.
> >
> > I also need some help interpreting Theorem 3 in Section 3.3. Theorem 3 seems to tell us that model reversal has an effect under LDP, but doesn’t tell us *what* the effect is (without knowing anything about the distribution $p_{\epsilon}(r)$). Even if it is beyond the scope of the paper to analyze these distributions, perhaps it would be possible to describe some general behaviors of $p_{\epsilon}(r)$ that would make model reversal more effective?

---

> ### Author Response · Authors · 2023-11-21
> **Response to Reviewer D9bv**
>
> Thank you for acknowledging the revisions made in our paper and for expressing satisfaction that we have addressed most of your concerns. Below are our responses to your new comments, and we hope these will further satisfy your queries and considerations.
>
> > I do share Reviewer sTQT’s sentiment that the techniques proposed in the paper aren’t tailored specifically to the problem setting (**of functional data and LDP**). For example, I agree that model reversal seems advantageous in the LDP setting (...). But it’s still not clear to me how or if the functional data setting is relevant to the proposed algorithms.
>
> We understand your concern. As mentioned in our response to Reviewer sTQT, our newly proposed techniques, Model Reversal (MR) and Model Averaging (MA), are not exclusively tailored for functional data. However, they are indeed specifically designed for LDP scenarios, and suitable for scenarios that necessitate the improvement and combination of weak classifiers. They have demonstrated good performance in both experiments and real application.
>
> > I also need some help interpreting Theorem 3 in Section 3.3. Theorem 3 seems to tell us that model reversal has an effect under LDP, but doesn’t tell us *what* the effect is (without knowing anything about the distribution $p_{\epsilon}(r)$).
>
> We apologize for any lack of clarity in Theorem 3. We plan to update Theorem 3 as follows to address this issue. If our explanation is still not clear enough or if you have any suggestions for further modifications, please let us know.
>
> **Theorem 3**. For a classifier $f_\varepsilon$ that adheres to $\varepsilon$-LDP, let's denote its classification accuracy rate as $r(x)=P(\text{sign}(f_\varepsilon(x))=y|x)$. Additionally, let $r_\delta$ represent the potential enhancement in classification accuracy that could be achieved for $f_\varepsilon$ through the application of model reversal.
>
> 1. If $r(x)=r_0$ for all $x(t)$, then $r_\delta=\max\lbrace1-2r_0,0\rbrace$.
>
> 2. Otherwise, let's denote the distribution of classification accuracy rate as $p_\varepsilon(r)=P(x(t) \in A_{\varepsilon,r})$ with $r\in[0,1]$, where $A_{\varepsilon,r}=\lbrace x(t)| r(x)=r \rbrace$. Then
>    $$
>    \mathbb{E}(r_\delta)=\int_0^1 p_\varepsilon(r)\max\lbrace 1-2r,0\rbrace dr=\int_0^{1/2} p_\varepsilon(r)(1-2r) dr.
>    $$
>
> Theorem 3 quantifies the enhancement in classification accuracy that model reversal can contribute to a classifier. We would also like to emphasize the following two points:
>
> 1. In practical scenarios, even if we have no information about the distribution $p_\varepsilon(r)$, employing MR will not worsen the results. It will either improve them or, at least, maintain them at their current level;
>
> 2. When $\varepsilon$ is sufficiently large, it represents a scenario with no LDP constraints. This implies that this theorem is also applicable in non-LDP contexts.
>
>
>
> > Even if it is beyond the scope of the paper to analyze these distributions, perhaps it would be possible to describe some general behaviors of $p_{\epsilon}(r)$ that would make model reversal more effective?
>
> When a classifier has a lower classification accuracy, meaning the distribution $p_\varepsilon(r)$ is more concentrated in the range $r \in [0, 1/2)$, using model reversal can bring about a more significant improvement.
>
> The four types of classifiers presented in our paper demonstrate differences in their distributions. We will update our paper soon to showcase the distributions of classification accuracy rates $p_\varepsilon(r)$ for different classifiers. In the meantime, you may refer to **Figure 1 in Section 5**, which includes error bars of the misclassification rate, and **Figure 3 in Appendix A.2**, which shows the boxplots, of various types of weak classifiers. These classifiers exhibit varying sensitivities to noise and demonstrate different distributions. For example, in **Figure 3**, the misclassification rate of the CG classifier remains highly volatile at smaller $\varepsilon$ values, ranging between $10$% and $90$%, instead of converging around $50$% as other methods do. This indicates that its performance can be significantly enhanced through model reversal.
>
> We appreciate your ongoing engagement in this discussion. Should there be any further points needing clarification or if you have additional questions, please do not hesitate to bring them up. Any additional feedback for enhancing our paper is highly valued. We are hopeful that these efforts will positively reflect in the paper's evaluation.

---

### Official Review · Reviewer_rZaz · 2023-11-01

**Soundness:** 3 good
**Presentation:** 2 fair
**Contribution:** 2 fair
**Rating:** 6
**Confidence:** 2

**Summary:**

This paper studies functional classification under local DP constraints. Specifically, the paper proposes a privatization step for each client using a projection-based dimensionality reduction technique. Then this paper also proposes a model average and a model reversal method to enhance the performance of the classifier. Finally, this paper also studies the same problem under heterogeneous multi-server setting.

**Strengths:**

1. This paper is the first to study functional classification with LDP constraints.
2. The proposed method is simple and practical.

**Weaknesses:**

1. It would be better if the authors could add some intuition and discussion about the result in Theorem 2 and 3.
2. The proposed method is built on the dimension reduction scheme and laplace mechanism which does not seem to be very novel.

**Questions:**

Please see weaknesses

---

> ### Author Response · Authors · 2023-11-20
> **Response to Reviewer rZaz**
>
> We thank the reviewer for the positive evaluation of our paper and the recognition of our method. Below, we do our best to address the reviewer's questions adequately.
>
>
>
> ### W1) It would be better if the authors could add some intuition and discussion about the result in Theorem 2 and 3.
>
> Thanks for the kind advice. We have now incorporated discussions in the revised manuscript.
>
> - In **Theorem 2**, we demonstrated that the validation process adheres to Local Differential Privacy (LDP). We then provided an unbiased estimate of the classifier's accuracy rate and the variance of this estimate. The accurate assessment of the classifier's performance is a fundamental prerequisite for the subsequent enhancement of the classifier's performance through  Model Reversal and Model Average (MRMA).
> - We have enriched the content of **Theorem 3**. This theorem measures the improvement in classification accuracy that can be achieved through Model Reversal (MR) when the distribution of classification accuracy rate of a classifier, constructed under $\varepsilon$-LDP, is known to be $p_\varepsilon(r)$ with $r\in[0,1]$.
>
>
>
> ### W2) The proposed method is built on the dimension reduction scheme and laplace mechanism which does not seem to be very novel.
>
> We apologize for not sufficiently highlighting the novelty of our work in the original submission. Indeed, the dimension reduction scheme and the Laplace mechanism are well-established methods. However, our paper focuses more on building a functional classifier and proposing the MRMA algorithm tailored for LDP to enhance classifier performance under LDP.
>
> For a detailed overview of the novelty of our work, we refer to **General response** for a discussion.
>
>
>
> Once again, we deeply appreciate the reviewer’s time for reviewing our manuscript. We are keen to continue the conversation and address any remaining points of confusion.

---

> ### Author Response · Authors · 2023-11-22
> **Seeking Feedback on Our Responses and Revised Submission**
>
> Dear Reviewer rZaz,
>
> We would like to thank you again for the precious time and constructive comments. In response to your comments, we have submitted a revised manuscript and would greatly appreciate any further thoughts or feedback you might have on these updates.
>
> As the discussion deadline is approaching, we are eager to address any remaining concerns or questions you may have regarding our revised submission. We deeply appreciate your advice and look forward to your prompt feedback.
>
> Best regards!

---

### Official Review · Reviewer_sTQT · 2023-11-01

**Soundness:** 3 good
**Presentation:** 3 good
**Contribution:** 2 fair
**Rating:** 6
**Confidence:** 3

**Summary:**

This paper studies the problem of (binary) classification for functional data, under local differential privacy.

Their approach is as follows: First they project the functional data onto a finite subspace and express it in a finite basis. This would make the data look like a point in $\mathbb{R}^d$. In order to reduce sensitivity, they apply either the $\tanh$ or the $\min$-$\max$ transformation to each point to make each coordinate bounded. This would give us bounded sensitivity. After that they train a classifier on the privatized transformation of the data.

In order to mitigate the effect of the noise they use a boosting technique, specifically they use model averaging: they divide the users into two groups, one for training and one for validation, and divide each of those two into $B$ parts. Then they design $B$ weak classifiers, one for each part of the training set and the evaluate them on the corresponding part from the validation set. Each validation user then outputs the result of the validation, with a randomized response mechanism to ensure privacy. If the answer from the aggregation of randomized response is less than 50% accurate, we negate the sign of the classifier to obtain a better classifier (Model Reversal). In order to design a final classifier using the $B$ weak classifiers and the validation outputs, they use the validation outputs to design a set of weights, and then to create the final classifier, they take a weighted average of the $B$ weak classifiers (Model Averaging).

They also provide a similar technique for combining the results of $K$ servers.

They run several experiments to compare the performance of their model against the naive model that uses all of the training data to train a single classifier and compare different threshold levels for Model Averaging.

**Strengths:**

I think this is the first paper that considers the problem of functional classification under local differential privacy.

From their experiments it seems like the accuracy of their approach is better than the naive baselines by 10 to 20 percent. The naive baselines include: all of the data is used to train one model, or we're doing model averaging but all of the weights are equal.

**Weaknesses:**

My main concern is limited novelty in the techniques. Apart from the part initial dimensionality reduction and encoding part, which are standard techniques, I think the rest of the techniques are independent of the functional classification setting specifically. Overall, it seems like once we're done with dimensionality reduction and encoding everything is the same as in $\mathbb{R}^d$.

The effect of model reversal in the experiments seems pretty small.

**Questions:**

It is mentioned that the ignored residual function contributes to privacy protection. I think that requires more explanation.

I think the budgeting for privacy at the top of page 4 is wrong. When running the Laplace mechanism and assuming sensitivity is some $\Delta$ you need to sample noise from $\mathcal{L}(\frac{\Delta}{\epsilon})^{\otimes d}$. You don't need to add a multiplicative $d$ for each dimension.

---

> ### Author Response · Authors · 2023-11-20
> **Response to Reviewer sTQT (part 1/2)**
>
> We thank the reviewer for the thorough review and excellent summarization of our work. We have carefully addressed each comment below and outlined the respective modifications made to our manuscript in light of the reviewer's feedback.
>
>
> ### W1) My main concern is limited novelty in the techniques.
>
> > Apart from the part initial dimensionality reduction and encoding part, which are standard techniques, I think the rest of the techniques are independent of the functional classification setting specifically. Overall, it seems like once we're done with dimensionality reduction and encoding everything is the same as in $R^d$.
>
> We thank the reviewer for the detailed and insightful comments. Dimensionality reduction and encoding are indeed standard techniques, while our paper is the first to integrate them into an algorithm for functional classification under Local Differential Privacy (LDP), where we proposed the MRMA algorithm tailored for LDP. And in our revised manuscript, we provided a detailed discussion about functional projection in **Section 3.1**. Additionally, in **Appendix B.1**, we offered further theoretical analysis on projection-based functional classification and measure the information loss induced by projection.
>
> Regarding "the rest of the techniques", we would like to emphasize the following points:
>
> - **Model Reversal (MR)**: This is a new technique we developed to enhance the performance of weak classifiers under LDP. In **Theorem 3 of Section 3.3**, we measured the improvement that model reversal can bring to a classifier;
> - **Model Average (MA)**: In our paper, we introduced a MA algorithm tailored for LDP. It includes our methods of evaluating the performance of weak classifiers under LDP (see **Theorem 2 of Section 3.3**), and assigning more suitable weights to these classifiers based on the evaluation;
> - **Regarding the Validation Process**: We advocate for allocating a larger proportion of clients to evaluate the performance of weak learners, especially when dealing with multivariate observational data under LDP. Since the evaluation is typically scalar, the noised client's evaluation of a model usually contains less noise than his noised observational vector under the same privacy protection level;
> - **Experimental and Real Data Results**: In Sections 5 and 6, our techniques, MR and MA, have significantly improved the performance of weak classifiers, especially in scenarios with large noise interference. Furthermore, the experimental results in Appendices A.3 and A.4 also validate our assertions regarding sample allocation.
>
> For a detailed overview of the novelty of our work, we refer to **General response** for a discussion.
>
>
> ### W2) The effect of model reversal in the experiments seems pretty small.
>
> We appreciate the feedback and apologize for any confusion caused by the presentation of our images. We have made revisions to the original figures and included additional simulations to better illustrate the improvements in classifier performance brought about by model reversal.
>
> Specifically, in **Figure 1 of Section 5**, we demonstrated the effectiveness of MR by comparing classifiers' performance with and without MR, where MR significantly improves the performance of all types of weak classifiers. And compare to MA, MRMA further enhances the performance of SVM and CG classifiers substantially. Similar results can also be found in **Fugure 2 in Section 6** (Real Application) and **Figure 7 in Appendix A.4**.

---

> ### Author Response · Authors · 2023-11-20
> **Response to Reviewer sTQT (part 2/2)**
>
> ### Q1) It is mentioned that the ignored residual function contributes to privacy protection. I think that requires more explanation.
>
> We thank the reviewer for the kind advice. We have added a more detailed discussion about finite basis projection in **Section 3.1** of the revised manuscript. Additionally, in **Figure 8 Appendix A.5**, we illustrated the impact of projection and transformation on functional observations using a real dataset.
>
> - As we discussed in **Section 3.1**, under LDP, compared to recovering true observational data, the primary concern is the performance of models trained on noised data. Utilizing finite basis projection is advantageous for capturing population-level pattern information, rather than individual differences. Ignoring the individual residual function, which contains personal information, does not significantly impact classifier performance while still effectively protecting user privacy.
> - **Figure 8 in Appendix A.5** visualizes functional observations from a phonemes dataset analyzed in Section 6. It also shows the curves reconstructed after finite basis projection and transformation. The projection (and transformation) erases individual detail information but retains the fluctuating pattern of individual curves, and the pattern of population-level group differences is also preserved.
>
>
>
> ### Q2) The budgeting for privacy at the top of page 4 is wrong.
>
> We apologize for the possible confusion. Maybe there was a misunderstanding in our original manuscript regarding the sensitivity $\Delta$. To clarify, $\Delta$ refers to the sensitivity of each element of the vector $\boldsymbol{z}^*$, i.e., $z_k^*$ , not the sensitivity of the vector $\boldsymbol{z}^*$. Therefore, according to the sequential composition theorem, to achieve $\varepsilon$-LDP for a $d$-dimensional vector, it is necessary to ensure that each dimension complies with $\varepsilon/d$-LDP. This underlines one of the challenges of handling high-dimensional data under LDP constraints.
>
>
>
> Once again, we appreciate the reviewer's precious time. We are eager to engage in further discussions to clear out any confusion.

---

> ### Comment · Reviewer_sTQT · 2023-11-21
>
> I have reviewed the responses and the updates in the paper. I thank the authors for their response. I will maintain my score for the time being.
>
> **W1**. My main criticism was that the techniques presented here, don't seem to be tailored to the setting of the problem, which is functional classification. After dimensionality reduction and encoding, the rest of the techniques apply to any data in $[-1,1]^d$. I don't think the authors have addressed this concern.
>
> **W2**. I have reviewed the new plots. I think comparing to the average misclassification rate of the weak classifiers as a baseline can be misleading. The authors should compare their results with the setting that there's no boosting style technique used. In their example $50$ weak classifiers are trained using $500$ of the $3000$ total clients. And then a final model is selected based on verification results that use on the weak classifier, using the remaining $2500$ clients. How would your models perform if you used the $3000$ clients on training a single classifier and assessed its performance? Same for the setting where real data is used.
>
> **Q1**. I understand the intuition that if we're summarizing the data, privatizing it should be easier. That being said, I wonder if there's any formalization of this claim.
>
> **Q2**. I see, I thought your goal is to privatize $\mathbf{z}^*$. In that case the $\ell_1$ sensitivity would have been $2d$. The final result is the same.

---

> > ### Author Response · Authors · 2023-11-21
> > **Response to Reviewer sTQT**
> >
> > Thank you for your additional comments. Your suggestions are important to us and highly appreciated.
> >
> >
> >
> > **W1.** We apologize for not clearly outlining the relationship among functional data, LDP, and our Model Reversal (MR) and Model Averaging (MA) techniques.
> >
> > You are correct in your observation; our newly proposed techniques, MR and MA, are not exclusively tailored for functional data. Actually, they are specifically designed for LDP scenarios. Our innovative MR technique aims to enhance weak classifiers, while our refined Model Averaging algorithm incorporates weight estimation tailored for LDP to ensemble these classifiers effectively. Regarding functional classification, we have other ongoing work exploring methods based on the general projection function. This paper does not delve deeper into that topic.
> >
> > The MR and MA techniques extend beyond functional data and are applicable to a variety of data types. We believe this adaptability is a strength of the MRMA algorithm, making it highly suitable for scenarios that necessitate the improvement and combination of weak classifiers. Moreover, their effectiveness is demonstrated through robust performance in our experimental results.
> >
> >
> >
> > **W2.** Thank you very much for your suggestion regarding the selection of baselines.
> >
> > In fact, in our original manuscript, we presented the results of classifiers that were not enhanced with any boosting-style techniques, which aligns with the method you suggested, namely those trained with all the available data. Compared to this classifier, the one based on our MRMA algorithm shows significantly better performance, especially in scenarios with large noise interference.
> >
> > When revising our paper, we incorrectly assumed that comparing our results with classifiers using no boosting-style technique might be unfair, as well-known ensemble models might perform better. Therefore, we removed the results of this classifier and added that of the classifiers based on classic aggregation methods. We appreciate you pointing this out, and we will promptly update our manuscript to include these results for a more comprehensive comparison.
> >
> >
> >
> > **Q1.** Wow, that's an excellent question, one that we hadn't considered in terms of theoretically studying how ignoring the residual function contributes to privacy protection. This idea might yield some new results that differ from the existing LDP framework. So far, our paper has only touched upon the impact of ignoring the residual function on classifier performance. However, as detailed in **Appendix B.1**, we measured the information loss caused by ignoring the residual function, which we believe can establish a starting point for future research into how ignoring the residual function contributes to privacy protection. We are very grateful for you bringing this up.
> >
> >
> >
> > **Q2.** We agree with your assessment. Again, we are very grateful for your careful reading and thoughtful feedback on our paper.
> >
> >
> >
> > Thank you for continuing the discussion with us. If there are any other aspects that we have not explained clearly, please feel free to raise them. We also warmly welcome any further suggestions for improvement.  We hope that this paper can get a better score.

---

> > > ### Comment · Reviewer_sTQT · 2023-11-22
> > >
> > > I thank the authors for their response.
> > >
> > > **W1.** I see, thank you.
> > >
> > > **W2.** Thank you for adding the all data accuracy plots. I'm curious, it's a bit surprising that the average model trained on $50$ samples has only slightly worse performance than a model trained on $3000$ samples. For example that is the case in Figure 1 (a), at $\varepsilon = 2$. Is there a good explanation for that? Also the behavior of the accuracy curve seems to differ significantly across different models. In figure 1, logistic and DWD, start deviating from $\varepsilon = 1$, while SVM and CG start deviating from $\varepsilon = 0.1$, is there an explanation for that behavior?
> > >
> > > **Q1.** I see. I think the wording in the paper makes it sound like we have strong evidence to suggest that ignoring the residual function would help with privacy. I suggest at least changing the wording, or alternatively providing evidence.
> > >
> > > > thereby enhancing privacy protection as the residual function ξ(t), containing personal information, is ignored.

---

> > > > ### Author Response · Authors · 2023-11-22
> > > > **Response to Reviewer sTQT**
> > > >
> > > > Thank you for continuing to engage with us and for providing crucial suggestions. We will make the corresponding revisions in our manuscript.
> > > >
> > > > **W2.** Thank you for raising questions about the experimental results. We will include additional explanations about the results in the paper.
> > > >
> > > > > I'm curious, it's a bit surprising that the average model trained on $50$ samples has only slightly worse performance than a model trained on 3000 samples. For example that is the case in Figure 1 (a), at $\varepsilon=2$. Is there a good explanation for that?
> > > >
> > > > The issue you've highlighted is indeed critical, and I believe it is a common challenge in privacy learning under LDP. Specifically, at lower values of $\varepsilon$ (indicating higher levels of privacy protection), the substantial noise interference can lead to a scenario where simply increasing the sample size for model training may not significantly improve model performance. This is what happened to Figure 1 when $\varepsilon\le2$.
> > > >
> > > > Furthermore, please consider reviewing **Figure 4 in Appendix A.2**. This figure displays the empirical density of classification accuracy rates for classifiers trained with $3000$ samples. It demonstrates that when $\varepsilon\le2$, classifier accuracy tends to converge around $50$% due to noise interference. And inspired by your comment, we will include a red vertical-line of mean value in each density plot for a more clearer comparison.
> > > >
> > > > Therefore, given these results, the MR (Model Reversal) and MA (Model Averaging) techniques we proposed specifically designed for LDP are both essential and significant.
> > > >
> > > > > Also the behavior of the accuracy curve seems to differ significantly across different models. In figure 1, logistic and DWD, start deviating from $\varepsilon=1$, while SVM and CG start deviating from $\varepsilon=0.1$, is there an explanation for that behavior?
> > > >
> > > > Your observation is perceptive, and we will address this with Theorem 3 in Section 3.3 and Figure 4 in Appendix A.2.
> > > >
> > > > **Theorem 3** measures the enhancement in classification accuracy that Model Reversal can contribute to a classifier, which depends on the privacy budget $\varepsilon$ and the distribution $p_\varepsilon(r)$. Different types of classifiers have various shapes of distribution, resulting in different levels of improvement through Model Reversal.
> > > >
> > > > From **Figure 4 in Appendix A.2**, as $\varepsilon$ decreases, the distributions $p_\varepsilon(r)$ for logistic and DWD classifiers gradually converge around $0.5$. However, SVM maintains a small probability of achieving accuracy far from $0.5$ (random guess), particularly CG which has a substantial probability of this. Therefore, there is potential for SVM and CG weak classifiers to be improved through Model Reversal. Furthermore, when ensembling weak classifiers, Model Averaging can bring greater enhancement to SVM and CG weak classifiers.
> > > >
> > > > Additionally, we would like to explain why, in Figure 1, at $\varepsilon=0.1$, the MR and MA techniques did not bring a more significant improvement to the CG weak classifier. At this point, due to noise interference, the evaluation of weak classifiers based on only $50$ samples is not sufficiently accurate. According to **Theorem 2 in Section 3.3**, as we increase the number of clients allocated for validation, we can obtain a more accurate assessment, thereby selecting better weak classifiers and achieving a more effective ensembled classifier. And the experimental results in **Appendix A.4** also confirm this point. This rationale underpins our proposal to allocate a larger proportion of clients to evaluate the performance of weak classifiers instead of training them.
> > > >
> > > >
> > > > **Q1.** The issue you've raised is indeed valid, and we thank you for your thorough review of our paper. We will make modifications to the manuscript for more accurate expression.
> > > >
> > > > We are grateful for your ongoing feedback. Should you have any additional recommendations for modifications or if there are any aspects of our paper that require further clarification, please feel free to inform us.

---

> > > > > ### Comment · Reviewer_sTQT · 2023-11-23
> > > > >
> > > > > Thanks for the explanation. I think now I understand why there is not much improvement in accuracy in the high sample setting.
> > > > >
> > > > > I think the authors have addressed most of my concerns regarding the experiments. I will slightly increase my score. That being said, I still think the presentation of this work as an algorithm for classification of functional data is not accurate.

---

> > > > > > ### Author Response · Authors · 2023-11-23
> > > > > > **Appreciation for Reviewer sTQT**
> > > > > >
> > > > > > Thank you for your ongoing thoughtful feedback, which has significantly helped us improve our manuscript, and for reconsidering your score. We are pleased to hear that we have addressed most of your concerns regarding the experiments.
> > > > > >
> > > > > > We understand your latest concern. In statistics, it is a common practice to reduce the dimensionality of functional data through projection and subsequently model based on these lower-dimensional projections, as detailed in the papers in "Functional Data Projection" in Section 2. Our paper adopts this approach and then shifts focus towards enhancing the performance of models learned under LDP, which is a more general and significant issue in privacy learning under LDP.
> > > > > >
> > > > > > Regardless, we are deeply grateful for all your previous feedback and advice, which have been invaluable to us.

---

### Author Response · Authors · 2023-11-20
**General Response to Reviewer Comments on Novelty**

We appreciate the reviewers for their constructive comments, which have significantly enhanced our manuscript. While the reviewers appreciate our pioneering and innovative approach in functional data classification under LDP, noting its effectiveness in enhancing classifier accuracy, their major concern comes from the paper's novelty (Reviewer sTQT, rZaz, and WLyT). We apologize for any shortcomings in our writing that may have obscured the presentation of our contributions and novelty, and we appreciate this opportunity to restate them here.

In summary, our paper is a pioneer in functional classification under LDP, introducing a projection-based classifier and proposing novel techniques: Model Reversal to improve weak classifiers and a Model Average algorithm for ensembling weak classifiers designed for LDP. These advancements substantially enhance classification performance, particularly in environments with high noise, which corresponds to higher levels of privacy protection.

1. **Functional-Projection under LDP vs DP**:

   As reviewer WLyT pointed out, there are existing works about projecting data onto a finite basis within DP. However, DP focuses on releasing functional summaries with access to all the raw data, whereas LDP emphasizes learning models from collected noisy functional observations. These represent distinctly different challenges. In the context of LDP, privacy learning with functional data, through finite basis projection, effectively reduces dimensionality to lessen noise interference, while simultaneously ensuring less impact on the efficacy of the learned models.

2. **The first to consider functional classification under LDP**:

   Our paper is the first to consider functional classification under LDP. We offer **theoretical analysis in Appendix B.1** on projection-based functional classification and measure the information loss in classification induced by projection. **Lemma 1** reveals that, under certain conditions, classifiers based on projection with finite basis can achieve lowest possible misclassification rate among all possible classifiers.

3. **Introduce an innovative technique, Model Reversal (MR), under LDP**:

   MR is a novel technique we developed to improve the performance of weak classifiers under LDP by inverting their prediction direction (by $-1*$ coefficient estimate) when their accuracy is below a certain threshold. Given a classifier's accuracy rate distribution under $\varepsilon$-LDP, **Theorem 3 in Section 3.3** measures the improvement that MR can bring to a classifier.

4. **Propose a Model Average (MA) algorithm tailored for LDP**:

   In our paper, we introduce a MA algorithm tailored for LDP. It includes our idea of allocating a larger proportion of clients to evaluate the performance of weak classifiers instead of their training. It also builds upon our methods of evaluating the performance of weak classifiers under LDP (see **Theorem 2 in Section 3.3**), and assigning more suitable weights to these classifiers based on the evaluation.

We effectively integrate these methods to construct functional classifiers designed for both **single-server and heterogeneous multi-server** environments under LDP. In both **experiments and real data applications**, classifiers developed using our proposed techniques, model reversal and model average, demonstrate excellent performance, especially in scenarios with large noise interference.

---

### Author Response · Authors · 2023-11-20
**Responses and Updated Manuscript Submitted**

We appreciate the reviewers' acknowledgment of our work as the first one ($\textcolor{red}{\text{sTQT}},\textcolor{green}{\text{rZaz}}$) in the topic of functional data classification under LDP, which is a novel setting ($\textcolor{blue}{\text{D9bv}}$), and a topic that has not been extensively explored ($\textcolor{darkorchid}{\text{WLyT}}$). We are glad that they found our method simple, practical ($\textcolor{green}{\text{rZaz}}$), novel ($\textcolor{blue}{\text{D9bv}}$), and improving the accuracy of classifiers ($\textcolor{red}{\text{sTQT}},\textcolor{blue}{\text{D9bv}}$). We are encouraged that they found our idea of model reversal fun and creative ($\textcolor{blue}{\text{D9bv}}$), and particularly innovative ($\textcolor{darkorchid}{\text{WLyT}}$).

We found the constructive feedback of the reviewers very helpful and have prepared an updated version of our manuscript. Additions to the manuscript are marked in blue in the revised manuscript and summarized below. We reply to each reviewer in more detail in individual responses. In our revised manuscript:

- **Section 1 (Introduction)**: Inspired by $\textcolor{blue}{\text{D9bv}}$'s feedback, we added an introduction to functional data and, following $\textcolor{darkorchid}{\text{WLyT}}$'s comments, included a discussion on DP and LDP;
- **Section 2 (Related Work)**: We moved the literature on Supervised Learning Under LDP and Functional Data Projection from the initial introduction into this section and expanded it with a comparison of methods for functional data under DP and LDP, as suggested by $\textcolor{darkorchid}{\text{WLyT}}$;
- **Section 3 (Single Server)**: Addressing $\textcolor{red}{\text{sTQT}}$ and $\textcolor{darkorchid}{\text{WLyT}}$'s concerns, we included the discussion on finite basis projection in Section 3.1. In line with $\textcolor{green}{\text{rZaz}}$ and $\textcolor{darkorchid}{\text{WLyT}}$'s advice, we added discussion on Theorems 2 and 3 and revised the content of Theorem 3 in Section 3.3;
- **Section 5 (Experiments)**: Responding to $\textcolor{blue}{\text{D9bv}}$'s concern on the credibility of empirical evaluations, and $\textcolor{red}{\text{sTQT}}$, $\textcolor{darkorchid}{\text{WLyT}}$'s concerns in the effects of model reversal, we included results based on MR and comparisons with classic aggregation methods;
- **Section 6 (Real Application)**: We added this section in response to $\textcolor{blue}{\text{D9bv}}$ and $\textcolor{darkorchid}{\text{WLyT}}$'s suggestions;
- **Appendix**: Per $\textcolor{darkorchid}{\text{WLyT}}$'s comments, we added Section A.4 to show the results of weak classifiers with varying sample sizes; Addressing $\textcolor{red}{\text{sTQT}}$ and $\textcolor{darkorchid}{\text{WLyT}}$, we included Section A.5 to visualize functional observations in real applications, and introduced Lemma 1 in Section B.1, concerning projection-based classification; To accommodate the additions above, we relocated the experiments data generation process to Appendix A.1, and algorithm and experiment results on the multi-server with federated learning to Appendix A.6.


### Updated Manuscript Submitted (21 Nov)

Thanks to the reviewers for their further comments and insightful feedback. Following their valuable suggestions, we have extensively revised and enhanced our paper. In our revised manuscript:

- **Section 1 (Introduction)**: We have more accurately and clearly summarized the contributions of our paper.
- **Section 3 (Single Server)**: In response to reviewer $\textcolor{blue}{\text{D9bv}}$'s concerns, we have revised Theorem 3 for clearer expression.
- **Section 5 (Experiments) and Section 6 (Real Application)**: Following reviewer  $\textcolor{red}{\text{sTQT}}$'s suggestion, we have included results of classifiers trained with all clients in the experiments and real application section.
- **Appendix**: Inspired by reviewer $\textcolor{blue}{\text{D9bv}}$'s comments, we have added the empirical distribution of the classification accuracy rate $p_\varepsilon(r)$ of different classifiers in **Appendix A.2, Figure 4**, to facilitate a better understanding of the distribution $p_\varepsilon(r)$ mentioned in Theorem 3.

### Updated Manuscript Submitted (22 Nov)

Inspired by Reviewer $\textcolor{red}{\text{sTQT}}$'s comments and advice, we have added a paragraph discussing Theorem 2 in Section 3.3, revised the discussion on experimental results in Section 5, and enriched Figure 4 in Appendix A.2. We are very grateful for the reviewer's comments, which have helped us improve the presentation of our paper.

---

### Author Response · Authors · 2023-11-23
**Further Discussion**

Dear Reviewers,

We sincerely thank all reviewers for their constructive feedback during the review and subsequent discussion phases. Our revised manuscript, refined with the aid of your insightful comments, has been submitted.

As the discussion period nears its conclusion, we kindly invite reviewers rZaz and WLyT to confirm whether their concerns have been adequately addressed. We hope our revisions meet your expectations. Should you have any further questions or points of concern, we are fully prepared to address them in pursuit of improving our evaluation.

We deeply appreciate the time and effort all reviewers have dedicated to reviewing our work and contributing to its improvement.

Kind regards,

the Authors

---

### Author Response · Authors · 2023-11-23
**Final Request for Feedback**

Dear Reviewers,

With less than an hour remaining in the discussion period, we invite any final comments or suggestions you might have, or a reconsideration of your evaluation of our paper. Your insights and feedback are greatly appreciated, and we are thankful for the effort and time you have dedicated to reviewing our work.

Warm regards,

the Authors

---

### Meta-Review · Area_Chair_ooJa · 2023-12-06

**Metareview:**

This paper considers functional classification (FC) setting with local differential privacy (LDP). In FC, each sample is an infinite dimensional $x$ and the label is a boolean $y$. The goal is to find a function $f$ such that for classification, whereas the misclassification loss here is based on whether $sign(f(x))$ agrees with $y$. Here we want to do this with LDP, where each client $i$ has $(x_i, y_i)$ and we can only send privatize information to the server. The approach of this paper is through dimensionality reduction on $x_i$. Once we have finite dimension, noise is added to the samples and labels, and models (i.e. $f$'s) are trained based on SVM, logistic regression etc on this noisy samples. It turns out that this results in terrible utility with many of the models not even achieving 50% accuracy!! To fix this, the authors propose *model reversal (MR)* and *model averaging (MA)*. MR is to flip the sign of the models with <50% accuracy. MA is to take a linear combination of models. By employing these together, the authors achieve much improved empirical results.

## Strengths

- First paper to study Functional Classification Under Local Differential Privacy

- The *model reversal (MR)* idea (used in boosting the accuracy of weak models) seems novel; it is also surprising that it works very well empirically.

## Weaknesses

- MR is very counterintuitive and the paper does not explain well why this works. This technique (MR+MA) should be evaluated further. For example, it would be interesting to see what happens if we just take random functions and do MRMA on them (that is, we don't even try to train the weak classifiers at all). It should also be compared to other boosting techniques

- Limited theoretical results: Theorems 1-2 are standard in DP. Meanwhile, theorem 3 (for MR; added during rebuttal) is both trivial and non-rigorous. (Namely, it is not clear what "the potential enhancement in classification accuracy" means especially when we take linear combination of multiple models which then interfere with each other.)

- Writing: The writing does not do a good job as to specify why the techniques here are specific for functional classification and not classification in general.

- It is not well explained why the weak classifiers are trained as they are (i.e. adding noise to $z$ and also to $y$), given the resulting classifiers are very weak. For example, there have been works on LDP model training (e.g. [1]) but the authors never discuss or explain why this is not used. It is very plausible that the results seen in the paper (weak classifiers have >50% error) is only the case because the training is not adapted appropriately to the LDP setting.


### References
[1] On Sparse Linear Regression in the Local Differential Privacy Model. Di Wang, Jinhui Xu Proceedings of the 36th International Conference on Machine Learning, PMLR 97:6628-6637, 2019.

**Justification For Why Not Higher Score:**

I view the paper's only novel contribution to be the model reversal (MR) technique, as other components are standard tools in literature. MR itself is not well explained and thus I think the paper requires a significant revision to investigate the behavior of MR further. I don't think it can be published in the current state.

**Justification For Why Not Lower Score:**

N/A

---

### Decision · Program_Chairs · 2024-01-16

Reject